# The TSPO-NOX1 axis controls phagocyte-triggered pathological angiogenesis in the eye

Anne Wolf [1,4], Marc Herb [2,4], Michael Schramm [2] & Thomas Langmann [1,3✉]

Aberrant immune responses including reactive phagocytes are implicated in the etiology of age-related macular degeneration (AMD), a major cause of blindness in the elderly. The translocator protein (18 kDa) (TSPO) is described as a biomarker for reactive gliosis, but its biological functions in retinal diseases remain elusive. Here, we report that tamoxifen-induced conditional deletion of TSPO in resident microglia using Cx3cr1[CreERT2]:TSPO[fl/fl] mice or targeting the protein with the synthetic ligand XBD173 prevents reactivity of phagocytes in the laser-induced mouse model of neovascular AMD. Concomitantly, the subsequent neoangiogenesis and vascular leakage are prevented by TSPO knockout or XBD173 treatment. Using different NADPH oxidase-deficient mice, we show that TSPO is a key regulator of NOX1-dependent neurotoxic ROS production in the retina. These data define a distinct role for TSPO in retinal phagocyte reactivity and highlight the protein as a drug target for immunomodulatory and antioxidant therapies for AMD.

[1] Laboratory for Experimental Immunology of the Eye, Department of Ophthalmology, University of Cologne, Faculty of Medicine and University Hospital Cologne, D-50931 Cologne, Germany. [2] Institute for Medical Microbiology, Immunology and Hygiene, D-50931 Cologne, Germany. [3] Center for Molecular Medicine Cologne (CMMC), University of Cologne, D-50931 Cologne, Germany. [4] These authors contributed equally: Anne Wolf, Marc Herb. ✉email: thomas.langmann@uk-koeln.de

Age-related macular degeneration (AMD) is a multi-factorial retinal degenerative disease that leads to severe vision loss amongst the elderly. The number of AMD patients is expected to reach 196 million worldwide by 2020 and increase to 288 million by 2040[1]. Early AMD is characterized by drusen deposits in the subretinal region of the macula. Late-stage AMD can manifest either as geographic atrophy (GA) (dry form) or as the wet form characterized by choroidal neovascularization (CNV). Angiogenic growth factors including VEGF-A promote the formation of abnormal leaky blood vessels[2] and the treatment of neovascular AMD currently relies on intravitreal injections of VEGF inhibitors[3]. However, these anti-VEGF therapies have significant limitations such as the burden of frequent intravitreal injections and resistance to treatment[4]. Moreover, there is currently no approved treatment available for GA patients.

While the etiology of AMD is still not well understood, genome-wide association studies and experimental animal models have unequivocally shown dysregulated innate immune responses in AMD involving complement and mononuclear phagocytes, including local microglia[5,6]. In the healthy retina, microglia constitute a surveillance network in the plexiform layers to preserve tissue integrity[7]. After injury, they proliferate and migrate to the site of damage to phagocytose cell debris, releasing neuromodulatory factors to promote tissue repair[8]. While a short period of controlled activation is considered to be neuroprotective, chronic microglia reactivity may endanger the already compromised retinal tissue[9]. Therefore, the release of proinflammatory cytokines and reactive oxygen species (ROS) and the concomitant phagocytosis of healthy neurons can be a driving force for photoreceptor demise and disease progression[8,10,11].

We and others have previously discovered that the translocator protein (18 kDa) (TSPO), a biomarker of brain microgliosis, is highly induced in microglia of degenerating retinas[12,13]. TSPO, together with its Müller cell-derived endogenous ligand diazepam-binding inhibitor (DBI), regulates the magnitude of microglia responses[13]. Based on this concept of feedback regulation, synthetic TSPO ligands showed potent immunomodulatory and neuroprotective properties in acute light-induced retinal damage[14] and other models for Alzheimer's disease[15], multiple sclerosis (MS)[16], and peripheral nerve injury[17].

Despite the well-documented expression of TSPO in microgliosis, its actual molecular functions are controversially discussed and include a role in steroidogenesis[18], mitochondrial metabolism and quality control[19,20], and more general cellular processes, such as proliferation, survival, and apoptosis[21]. Recent studies established global and cell-specific TSPO knockout mice to challenge the proposed role of TSPO in regulation of the mitochondrial permeability transition pore and as a gate-keeper of steroid hormone biosynthesis[22,23].

In this study, we focus on elucidating the molecular function of TSPO in retinal immune homeostasis and angiogenesis. Using the laser-CNV model, an established system to study key aspects of neovascular AMD[24], we demonstrate that microglia-specific TSPO-KO and TSPO ligand treatment strongly diminish mononuclear phagocyte reactivity and neoangiogenesis. We show that neurotoxic ROS production in microglia is regulated by a TSPO-mediated increase in calcium levels and activation of NADPH oxidase 1 (NOX1). These findings highlight a distinct role for TSPO in ROS production of phagocytes and provides a molecular mechanism for TSPO/ROS-related immunomodulatory and neuroprotective therapies in the retina and the brain.

## Results

**XBD173 prevents phagocyte reactivity in laser-induced CNV.** We have previously shown that microglia activation and light-induced retinal degeneration can be prevented by TSPO ligands[14]. To investigate if targeting TSPO with the ligand XBD173 has also immunomodulatory potential in an in vivo model of neovascular AMD, we performed laser photocoagulation in C57BL/6J mice that were treated with daily intraperitoneal injections of XBD173 or vehicle. We first monitored the immune-related effects of XBD173 in laser-induced CNV. Confocal images of retinal flat mounts from DMSO-treated mice revealed massive accumulation of reactive ameboid-shaped Iba1$^+$ cells within the lesions at 3 days post-laser injury, whereas retinas from XBD173-treated mice had less Iba1$^+$ phagocytes and these cells showed mainly a ramified morphology (Fig. 1a, b). The infiltration of immune cells in the retina 7 days after laser injury was overall less than at 3 days, indicating a wound healing process. Nevertheless, XBD173 treatment significantly attenuated phagocyte reactivity at both time points (Fig. 1b, c). The mRNA expression of Tspo itself and Cd68 were then quantified to determine the magnitude of immune cell activation. Indeed, retinal Tspo and Cd68 transcript levels strongly increased after laser injury compared to naïve mice and the XBD173-treated groups showed diminished activation marker expression especially at the earlier time points (Supplementary Fig. 1a, b). TSPO protein oligomerization has been reported in human and mouse cells[25] and we therefore analyzed retinal TSPO levels under non-reducing conditions. After laser injury, Western blot analysis revealed a higher TSPO-specific molecular weight band at 25 kDa (referred to as HMW1), that was absent in non-lasered naïve or XBD173-treated mice (Fig. 1d). In contrast, monomeric TSPO levels (referred to as LMW), were significantly lower compared to naïve mice. The ratio of HMW1 to LMW was higher after laser injury than in naïve mice and XBD173 prevented lesion-associated formation of HMW1 TSPO (Fig. 1d, e). We next focused on the secretion of pro-inflammatory cytokines. Six hours after laser injury, CCL2 and IL-6 were found in the retinal tissue, whereas levels of IL-1β and TNF did not change (Fig. 1f). Notably, XBD173-injected mice had strongly reduced CCL2 and IL-6 secretion comparable to the level of naïve mice (Fig. 1f).

To explore the effects of TSPO targeting on subretinal and RPE-associated phagocytes, Iba1$^+$ cells were imaged in RPE/choroidal flat mounts. Similarly, as in the retina, XBD173 treatment reduced phagocyte infiltration and reactivity in the RPE/choroid compared to vehicle injections (Fig. 1g–i). Laser-induced Tspo and Cd68 expression were also reduced after XBD173 treatment (Supplementary Fig. 1a, b). Of note, Western blot analysis of RPE/choroid revealed an additional TSPO-specific HMW band (36 kDa) (referred to as "HMW2") (Fig. 1j). Again, LMW TSPO levels were significantly lower and the ratio of HMW1 to LMW and HMW2 to LMW was higher after laser-injury than in naïve mice and significantly reduced in XBD173-treated mice (Fig. 1k). Moreover, levels of CCL2, IL-6, and IL-1β increased in the RPE/choroid after laser-injury and XBD173 treatment prevented their laser-induced secretion (Fig. 1l).

Since reactive mononuclear phagocytes are a rich source for ROS, that have been suggested as drivers of neurodegeneration[26], we next analyzed if targeting TSPO with XBD173 also affects ROS production of mouse primary microglia in culture. We first analyzed extracellular and phagosomal ROS production, which can be measured with the cell-impermeable dye isoluminol[27]. These ROS strongly increased after stimulation of microglia with PMA or after phagocytosis of photoreceptor cell debris (Fig. 1m and Supplementary Fig. 2a). Culture of the microglia in the presence of XBD173 strongly diminished stimulation-induced ROS production (Fig. 1m). In addition, treatment with four other TSPO ligands, including Ro5-4864, PK11195, Etifoxine, and FGIN-1-27 also resulted in reduced stimulation-induced ROS production (Supplementary Fig. 3). In contrast, cytosolic ROS or

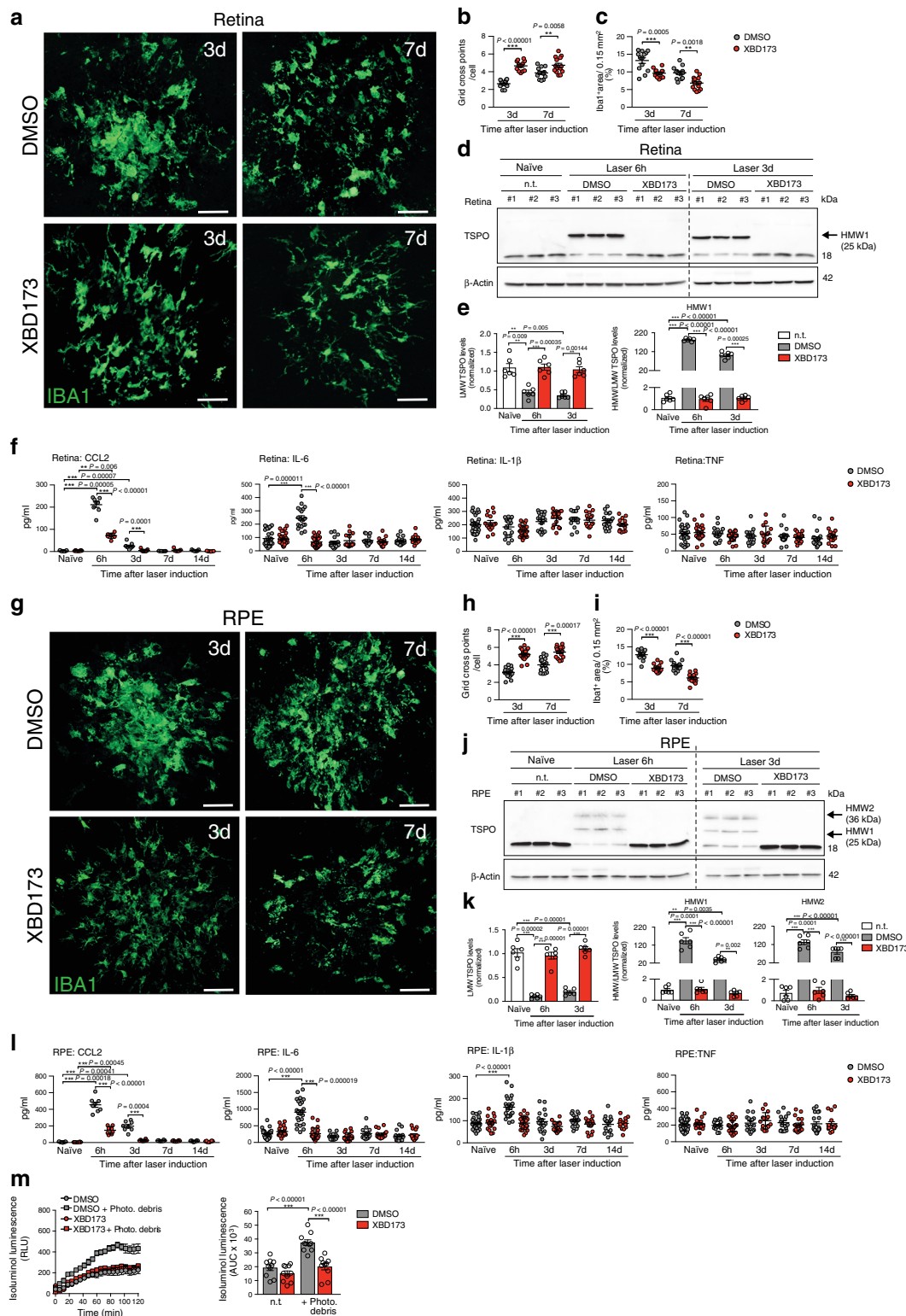

ROS produced in the mitochondrial matrix could not be detected in stimulated microglia (Supplementary Fig. 4a, b). These data indicate that the TSPO ligand XBD173 blocks extracellular and phagosomal ROS production of microglia.

**XBD173 limits laser-induced vascular leakage and CNV.** To investigate the anti-angiogenic potential of XBD173, we assessed its effects on inflammation-induced vascular leakage with late-phase fundus fluorescein angiography (FFA). While vehicle-injected mice showed prominent vascular leakage after laser injury, strongly reduced vascular leakage was seen in XBD173-treated mice at all analyzed time points (Fig. 2a). Both, leakage intensity and area were significantly lower in the XBD173 group than in controls (Fig. 2b, c). We confirmed these findings by monitoring CNV formation using lectin staining of RPE/choroidal flat mounts. The CNV size was significantly smaller in the XBD173 treatment groups compared to vehicle control mice (Fig. 2d, e). To elucidate whether

**Fig. 1 XBD173 dampens phagocyte reactivity in laser-CNV. a** Representative images of Iba1+ phagocytes within retinal laser lesions. Scale bar: 50 μm. **b** Quantification of Iba1+ cell morphology within lesions. DMSO/XBD173 n = 17/13 spots. **c** Quantification of Iba1+ area in lesions. n = 13 spots. **d** TSPO protein in retinas of naïve and lasered mice at indicated time points. Each lane represents an individual retina. Dotted line indicates individual blots processed in parallel. **e** Densitometry of western blots. LMW TSPO signals were normalized to β-Actin and HMW:LMW TSPO ratio determined. n = 6 retinas from two independent experiments. **f** Cytokines in retinas of naïve and lasered mice at indicated time points. CCL2 (n = 8 retinas from individual mice); IL-6 (naïve n = 32; 6 h n = 25; 3 d/7 d/14 d n = 17 retinas from individual mice), IL-1β (DMSO/XBD173 naïve n = 33/25; 6 h n = 25; 3 d, 7 d, and 14 d n = 17 retinas from individual mice) and TNF (DMSO/XBD173 naïve n = 33/24; 6 h, 3d, 7d, and 14d n = 17 retinas from individual mice). **g** Representative images of Iba1+ cells in RPE/choroidal laser lesions. Scale bar: 50 μm. **h** Quantification of Iba1+ cell morphology within lesions. **i** Quantification of Iba1+ area in laser lesions. n = 13 spots. **j** TSPO protein in RPE/choroids of naïve and lasered mice at the indicated time points. **k** Densitometry of Western blots. n = 6 RPE/choroids from two independent experiments. LMW lower molecular weight, HMW, higher molecular weight, n.t. non-treated. **l** Cytokine levels in RPE/choroids of naïve and lasered mice. CCL2 (n = 8 RPE/choroids from individual mice); IL-6, IL-1β, and TNF (DMSO/XBD173 naïve n = 30/20; 6 h n = 23; 3 d, 7 d, 14 d n = 17 RPE/choroids from individual mice. **m** Quantification of extracellular ROS production by primary microglia from wild type mice. Kinetics of ROS production and the area under the curve (AUC) are shown. n = 11 independent experiments. Data show mean ± SEM. ROS data were analyzed using two-tailed unpaired Student's t test. A linear mixed model was used for laser-CNV data; **P < 0.01 and ***P ≤ 0.001. Source data are provided as a Source Data file.

targeting of TSPO also affects angiogenic growth factors, protein levels of VEGF-A, ANG-1, ANG-2, and IGF-1 were measured in the retina and RPE/choroid. The secretion of all growth factors was significantly increased in both tissues after laser injury, but strongly reduced in XBD173-treated mice especially at early time points of the analysis (Fig. 2f, g). As laser-induced CNV is also accompanied by a wound healing process, we monitored the lesion size and the formation of a fibrotic scar over time after laser injury using SD-OCT. Notably, XBD173 treatment attenuated the lesion-associated fibrosis significantly at all time points compared to controls, indicating a faster wound healing process (Fig. 2h, i). These findings demonstrate that XBD173 attenuated both vascular leakage and neoangiogenesis.

**TSPO-KO blocks microglia reactivity and ROS production.** We next assessed the direct function of TSPO in retinal immune cells and generated microglia-specific conditional TSPO-KO mice (referred to as TSPO^ΔMG) by breeding TSPO^fl/fl mice with Cx3cr1^CreERT2 mice. This tamoxifen-inducible Cre-loxP system efficiently targets long-lived resident immune cells including microglia in the retina and brain[28]. Successful microglia-specific deletion of TSPO was confirmed by genomic PCR and Western blot analysis of retinal and RPE/choroidal lysates from tamoxifen-treated and TSPO^ΔMG and TSPO^fl/fl control mice (Supplementary Fig. 5a, b). We also validated the microglia-specific TSPO-KO in primary microglia isolated from TSPO^ΔMG and TSPO^fl/fl mice (Supplementary Fig. 5c, d). Notably, microglia from TSPO^ΔMG mice did not show obvious phenotypical differences in their ramification or branching network compared to TSPO^fl/fl mice (Supplementary Fig. 5e, f). Since TSPO may be involved in basic mitochondrial functions[19,20], we analyzed the mitochondrial network, the mitochondrial membrane potential (MMP) and cellular ATP levels (Supplementary Fig. 6). Confocal image analysis of the mitochondrial network in TSPO^ΔMG microglia showed no mitochondrial fragmentation or other alterations in morphology compared to TSPO^fl/fl microglia, where TSPO co-localized with mitochondria (Supplementary Fig. 6a). Interestingly, unstimulated TSPO^ΔMG microglia showed a slight hyper-polarization of MMP compared to TSPO^fl/fl cells, suggesting an increased activity of the electron transport chain (Supplementary Fig. 6b). However, after stimulated phagocytosis of photoreceptor debris, the MMP was slightly reduced in both TSPO^ΔMG and TSPO^fl/fl cells (Supplementary Fig. 6b). Microglia from TSPO^ΔMG mice showed no differences in cellular ATP levels compared to TSPO^fl/fl microglia neither in untreated conditions nor after stimulation with debris (Supplementary Fig. 6c). Inhibition of glycolysis through the glucose derivative 2-deoxy-D-glucose (2-DG) showed that both resting and stimulated microglia depend to

some degree on glycolysis for ATP generation (Supplementary Fig. 6d). However, inhibition of the mitochondrial ATP synthase by oligomycin A treatment showed that ATP is mainly generated through mitochondrial respiration (Supplementary Fig. 6e). Importantly, there was no difference between WT and TSPO knockout microglia. Thus, TSPO-deficient microglia are perfectly capable of generating ATP through mitochondrial respiration further indicating unimpaired mitochondrial function. Taken together, these data implicate that TSPO is not required for mitochondrial integrity, health, or energy metabolism in unstimulated or stimulated microglia.

We next determined if the absence of TSPO in microglia affects their behavior in the laser-CNV model. TSPO^ΔMG retinas showed less infiltration of Iba1+ cells and reduced reactivity in the laser lesions at all time points analyzed compared to TSPO^fl/fl mice (Fig. 3a–c). The increased expression of Tspo and Cd68 was also prevented in TSPO^ΔMG mice (Supplementary Fig. 1c, d). Notably, naïve TSPO^ΔMG mice had lower LMW TSPO levels in the retina than TSPO^fl/fl mice and the lesion-induced formation of HMW1 TSPO was only detected in TSPO^fl/fl but not in TSPO^ΔMG mice (Fig. 3d, e). Furthermore, the laser-induced secretion of CCL2 and IL-6 was prevented in TSPO^ΔMG retinas, phenocopying the effects of XBD173 treatment (Fig. 3f).

Similar to the retina, the analysis of phagocytes in RPE/choroidal flat mounts showed reduced Iba1+ cell infiltration and reactivity in TSPO^ΔMG mice compared to TSPO^fl/fl mice (Fig. 3g–i and Supplementary Fig. 1e, f). Also, laser-induced Tspo expression and formation of HMW1 and HMW2 TSPO were absent in the RPE/choroid of TSPO mice (Fig. 3j–k and Supplementary Fig. 1b). There was also missing CCL2, IL-6, and IL-1β secretion in the RPE/choroid after laser injury in TSPO^ΔMG mice (Fig. 3l).

When analyzing ROS production of microglia, we found that genomic deletion of TSPO specifically in these cells completely abolished their capacity to produce this neurotoxin after stimulation with photoreceptor cell debris (Fig. 3m and Supplementary Fig. 2b). Notably, XBD173 treatment could not further reduce ROS levels in TSPO^ΔMG microglia suggesting a TSPO-specific inhibitory effect on ROS production (Fig. 3m). No differences in cytosolic or ROS production into the mitochondrial matrix were found in TSPO^ΔMG versus TSPO^fl/fl microglia (Supplementary Fig. 4c, d).

**TSPO-KO in microglia prevents laser-induced CNV.** Since targeting of TSPO in microglia strongly reduced their inflammatory potential, we invested their effects on vascular leakage and CNV. TSPO^fl/fl mice showed typical laser-induced vascular leakage that was strongly reduced in TSPO^ΔMG mice at all

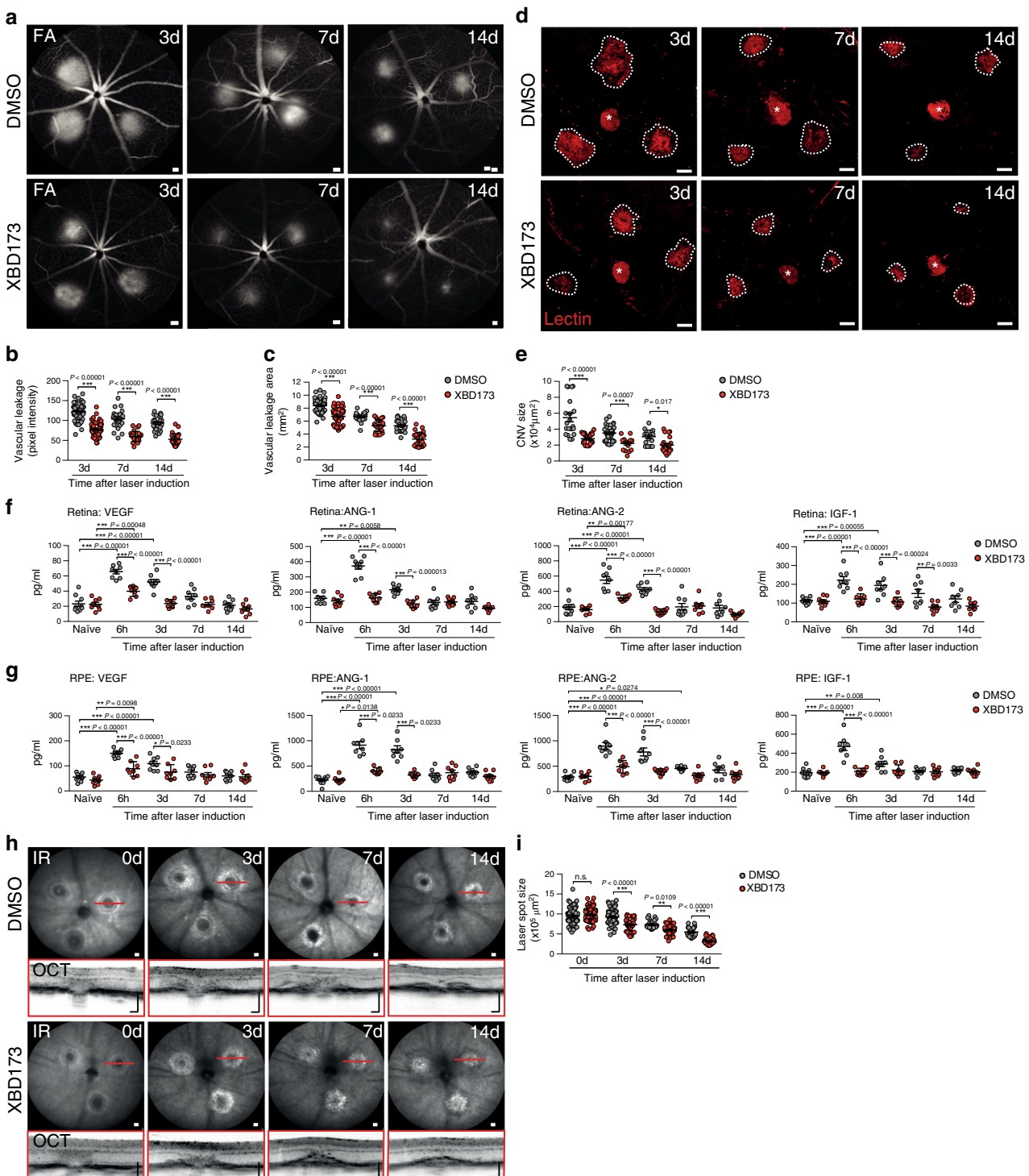

**Fig. 2 XBD173 inhibits laser-induced vascular leakage and pathological choroidal neovascularization (CNV) in mice. a** Representative late phase fundus fluorescein angiography (FFA) images at indicated time points post laser injury. Scale bar: 200 μm. **b** Quantification of vascular leakage intensity after laser-induced CNV. 3 d *n* = 56; 7 d, 14 d *n* = 25 eyes from individual mice. **c** Quantification of vascular leakage area after laser-induced CNV. 3 d *n* = 56; 7 d, 14 d *n* = 25 eyes from individual mice. **d** Representative laser-induced CNV stained with isolectin B4 in RPE/choroidal flat mounts. Scale bar: 100 μm; FA fluorescein angiography. **e** Quantification of laser-induced CNV area in RPE/choroidal flat mounts. DMSO/XBD173 3 d *n* = 18/34; 7 d *n* = 38/17; 14 d *n* = 17/23 RPE/choroids from individual mice. **f** Pro-angiogenic growth factor levels in retinas of naïve and lasered mice at indicated time points. *n* = 8 eyes from individual mice. **g** Pro-angiogenic growth factor levels in RPE/choroids of naïve and lasered mice at indicated time points. *n* = 8 eyes from individual mice. **h** Representative infrared (IR) fundus images at indicated time points post laser injury. Lower panel shows OCT scan from one laser spot marked by a red line. Scale bar: 200 μm. **i** Quantification of laser spot size. DMSO/XBD173 0 d *n* = 106/71; 3 d *n* = 72/42; 7 d, 14 d *n* = 26 eyes from individual mice. Data show mean ± SEM and a linear mixed model was used for statistical analyses; *P < 0.05; **P < 0.01; and ***P ≤ 0.001. n.s., not significant. Source data are provided as a Source Data file.

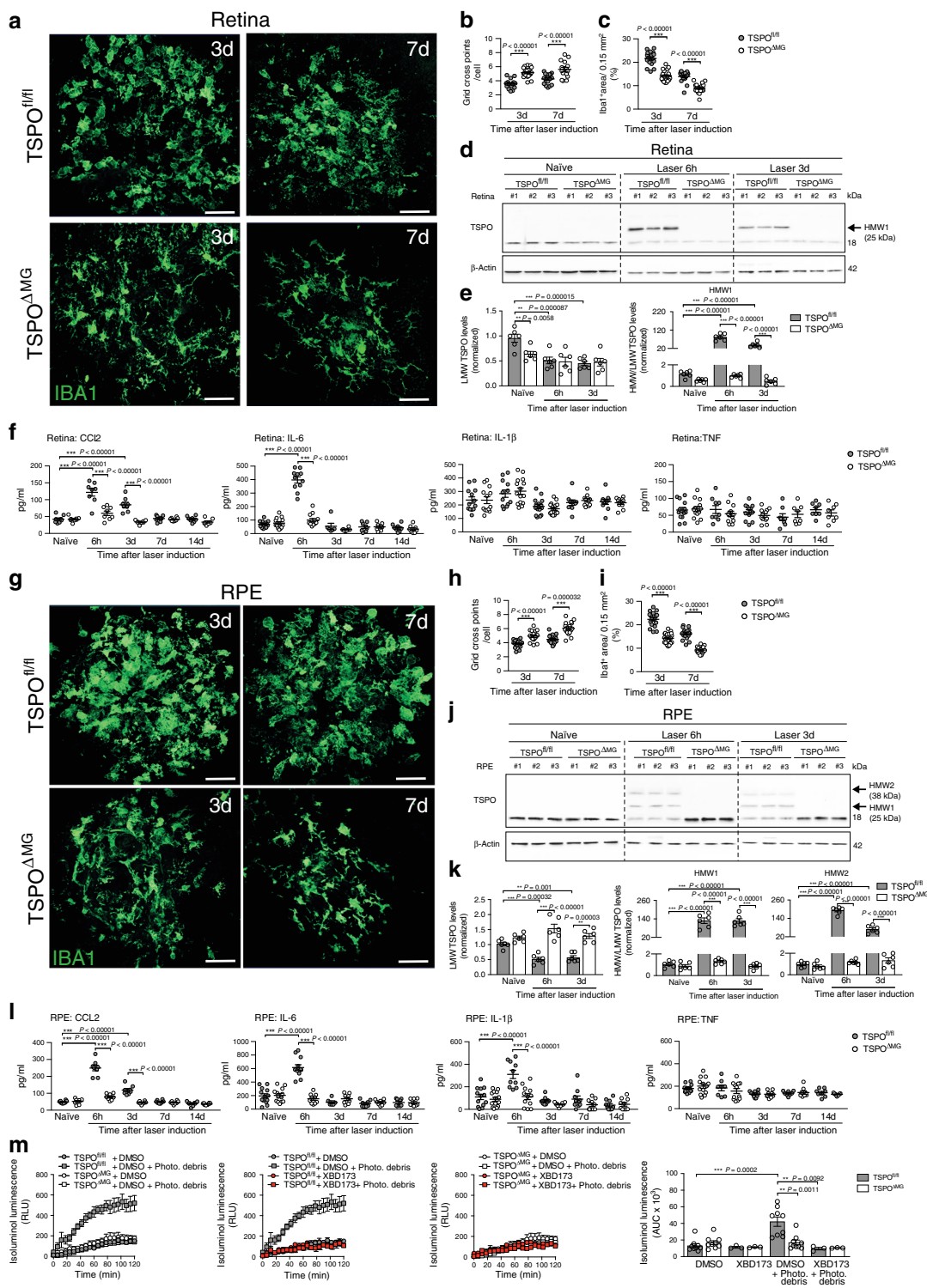

analyzed time points (Fig. 4a–c) and the overall CNV size was significantly smaller in TSPO$^{\Delta MG}$ mice than in controls (Fig. 4d, e). While a prominent laser-induced secretion of VEGF-A, ANG-1, ANG-2, and IGF-1 was found in the retina and RPE/choroid of TSPO$^{fl/fl}$ mice, only basal levels of these pro-angiogenic factors were detectable in TSPO$^{\Delta MG}$ mice (Fig. 4f–g).

Since XBD173 treatment and TSPO knockout in microglia significantly reduced the expression of these pro-angiogenic factors in vivo, we next analyzed whether microglia are the source of these factors. Analysis of in vivo transcribed *Vegf*, *Ang1*, and *Ang2* mRNAs by simultaneous in situ hybridization (ISH) of

RPE/choroidal flat mounts from TSPO$^{fl/fl}$ and TSPO$^{\Delta MG}$ mice with probes targeting *Aif* (*Iba1*) and *Vegf* showed that most *Iba1*+ mononuclear phagocytes expressed *Vegf* in the lesion area 3 days post laser injury (Supplementary Fig. 7a), while the overall expression of *Aif* and *Vegf* mRNA was reduced in TSPO$^{\Delta MG}$ mice as analyzed by the percentage of pixel intensity per image area (Supplementary Fig. 7b). Similar results were observed by simultaneous detection of *Aif/Ang1* (Supplementary Fig. 7c, d) and *Aif/Ang2* (Supplementary Fig. 7e, f). These results indicate that *Iba1*+ mononuclear phagocytes from TSPO$^{fl/fl}$ and TSPO$^{\Delta MG}$ mice express *Vegf*, *Ang1*, and *Ang2* mRNA after injury.

**Fig. 3 TSPO deficiency dampens phagocyte reactivity in laser-CNV. a** Representative images of Iba1+ cells within retinal laser lesions. Scale bar: 50 μm. **b** Quantification of Iba1+ cell morphology within lesions. $n = 18$ spots; TSPO$^{\Delta MG}$ 7 d $n = 21$ spots. **c** Quantification of Iba1+ area in lesions. $n = 18$ spots; TSPO$^{\Delta MG}$ 7 d $n = 21$ spots. **d** TSPO protein in retinas of naïve and lasered TSPO$^{fl/fl}$ and TSPO$^{\Delta MG}$ mice at indicated time points. Each lane represents an individual retina. Dotted line indicates individual blots processed in parallel. **e** Densitometry of western blots. LMW TSPO signals were normalized to β-Actin and HMW:LMW TSPO ratio determined. $n = 6$ retinas from two independent experiments. **f** Cytokines in retinas of naïve and lasered TSPO$^{fl/fl}$ and TSPO$^{\Delta MG}$ mice at indicated time points. CCL2 ($n = 8$ retinas from individual mice); IL-6, IL-1β, and TNF (naïve $n = 13$ retinas from individual mice); IL-6, TNF (3 d–14 d $n = 8$ retinas from individual mice) and IL-1β (6 h–14 d $n = 10$ retinas from individual mice). **g** Representative images of Iba1+ cells in RPE/choroidal lesions. Scale bar: 50 μm. **h** Quantification of Iba1+ cell morphology within lesions. $n = 18$ spots. **i** Quantification of Iba1+ area of the laser lesions. $n = 18$ spots. **j** TSPO protein in RPE/choroids of naïve and lasered TSPO$^{fl/fl}$ and TSPO$^{\Delta MG}$ mice at indicated time points. **k** Densitometry of Western blots. $n = 6$ RPE/choroids from two independent experiments. LMW lower molecular weight; HMW higher molecular weight. **l** Cytokines in RPE/choroids of naïve and lasered TSPO$^{fl/fl}$ and TSPO$^{\Delta MG}$ mice. CCL2 ($n = 8$ RPE/choroids from individual mice); IL-6 TSPO$^{fl/fl}$/TSPO$^{\Delta MG}$ 6 h $n = 11/12$, 3–14 d $n = 8$; IL-1β TSPO$^{fl/fl}$/TSPO$^{\Delta MG}$ 6 h $n = 10/12$, 3–14 d $n = 10$; TNF TSPO$^{fl/fl}$/TSPO$^{\Delta MG}$ 6 h $n = 9/12$, 3 d $n = 12$ and 7–14 d $n = 8$ RPE/choroids from individual mice. **m** Extracellular ROS production by microglia from TSPO$^{fl/fl}$ and TSPO$^{\Delta MG}$ mice. Kinetics and area under the curve (AUC) are shown. $n = 10$ independent experiments (+XBD173, $n = 3$ independent experiments). Data show mean ± SEM. ROS data were analyzed using two-tailed unpaired Student's $t$ test. A linear mixed model was used for laser-CNV data; $**P < 0.01$ and $***P \leq 0.001$. Source data are provided as a Source Data file.

We next analyzed the expression of these pro-angiogenic growth factors in isolated primary microglia (Supplementary Fig. 8). While *Ang1* and *Ang2* were only expressed at basal levels, *Vegf* expression was strongly increased after stimulation of microglia with photoreceptor cell debris. This increase was unaltered in either XBD173-treated or TSPO-deficient microglia (Supplementary Fig. 8a). Accordingly, only VEGF was secreted from stimulated microglia, which was unaltered in XBD173-treated or TSPO-deficient microglia (Supplementary Fig. 8b). These data show that stimulated microglia are capable of expressing and secreting VEGF independently of TSPO. Therefore, the strong reduction of pro-angiogenic growth factor levels observed in vivo after XBD173 treatment or in TSPO$^{\Delta MG}$ mice is very likely not due to TSPO-related pro-angiogenic growth factor secretion from microglia. Finally, we assessed the wound healing process in TSPO$^{\Delta MG}$ mice. These mice showed attenuated laser lesion sizes and significantly reduced fibrosis compared to TSPO$^{fl/fl}$ mice (Fig. 4h, i). Thus, TSPO knockout in retinal microglia considerably reduced inflammation-associated vascular leakage and neovascular lesions.

**TSPO triggers ROS production in microglia via NOX1.** Since our observations above indicated a role for TSPO in phagocyte ROS production, we further evaluated the molecular pathways. NADPH oxidases (NOX) are main producers of ROS, but mitochondria may also generate oxidants under certain conditions[27]. NOX2 often is considered as the major source of ROS in phagocytes[29]. However, the NOX family includes six other enzymes, NOX1, NOX3, NOX4, NOX5, and the dual oxidases DUOX1-2 that could be responsible for extracellular/phagosomal ROS production in microglia. Since NOX5 is absent in rodents[30], we first analyzed the expression pattern of the remaining NOX isotypes in the retina and RPE. In both tissues, all NOX family members except *Nox3*, were expressed at low basal levels (Supplementary Fig. 9a). Retinal laser injury did not induce expression of *Nox2*, *Nox4*, *Duox1*, or *Duox2* compared to naïve mice (Supplementary Fig. 9b, c). In contrast, *Nox1* expression strongly increased after laser-induced injury in the retina and RPE/choroid, which was abolished when the mice were treated with XBD173 (Fig. 5a). To validate these findings, extracellular microglial ROS production was determined in different Nox-deficient mice. Microglia deficient for p22$^{phox}$ (p22$^{phox}$-KO), the common catalytic subunit of NOX1-4, were lacking stimulation-induced extracellular ROS production, excluding DUOX1 and DUOX2 as ROS sources as both are not dependent on p22$^{phox}$ (Fig. 5b). Microglia from Nox2-deficient and Nox4-deficient mice were still able to produce stimulation-induced extracellular ROS, which was abolished after XBD173 stimulation (Supplementary

Fig. 10a, b). In contrast, Nox1-deficient microglia displayed no stimulation-induced ROS production (Fig. 5c) and no induced expression of *Nox1* was detected in TSPO$^{\Delta MG}$ mice after laser-CNV (Fig. 5d). These results indicate that NOX1 is the key enzyme for ROS production in retinal phagocytes and critically depends on the presence of TSPO.

**TSPO-associated increase in cytosolic Ca$^{2+}$ is essential for NOX1-derived ROS.** Calcium (Ca$^{2+}$) is an important second messenger that regulates a variety of cellular functions[31] and is responsible for the activation of ROS-generating enzymes[32]. To further investigate the functional coupling between TSPO and NOX1 in microglia, we first analyzed the effects of modulating Ca$^{2+}$ levels on ROS production. Increasing cytosolic Ca$^{2+}$ with the Ca$^{2+}$-ionophore ionomycin was sufficient to induce extracellular ROS production in non-stimulated microglia (Fig. 6a). Microglia from wildtype, TSPO$^{fl/fl}$, and TSPO$^{\Delta MG}$ mice completely failed to produce ROS in the absence of extracellular Ca$^{2+}$ (Fig. 6a, b). This indicates that the influx of extracellular Ca$^{2+}$ is essential for the induction of NOX1-dependent ROS production in response to stimulation with photoreceptor debris. Interestingly, not only stimulus-dependent NOX1 activity but also increased *Nox1* expression in primary microglia was detected after phagocytosis of photoreceptor debris, which was strongly reduced by XBD173 or in microglia-specific TSPO-KO (Supplementary Fig. 11). Notably, this up-regulation of *Nox1* gene expression was also depended on extracellular Ca$^{2+}$ levels (Supplementary Fig. 11), indicating that the increased ROS production is in part due to increased *Nox1* expression in primary microglia. Since mitochondria can serve as a Ca$^{2+}$ store[33], we next analyzed mitochondrial and cytosolic Ca$^{2+}$ levels of microglia and found that stimulated phagocytosis increased cytosolic but not mitochondrial Ca$^{2+}$ levels (Fig. 6c and Supplementary Fig. 12). Notably, XBD173 prevented the stimulation-induced increase in cytosolic Ca$^{2+}$ (Fig. 6c) as did the microglia-specific knockout of TSPO itself (Fig. 6d). These results demonstrate that the TSPO-dependent increase of cytosolic Ca$^{2+}$ after stimulation is essential not only for *Nox1* expression but also for stimulated NOX1-dependent ROS production.

**Nox1-KO impairs phagocyte reactivity in laser-induced CNV.** We next examined if the absence of NOX1 affects the phagocyte response in the laser-induced CNV mouse model. Immunostaining of retinal flat mounts from Nox1-KO mice showed reduced accumulation of Iba1+ cells within the laser lesion (Fig. 7a–c) albeit no morphological differences in their ramification was detected (Fig. 7b). Nox1-deficient mice also displayed

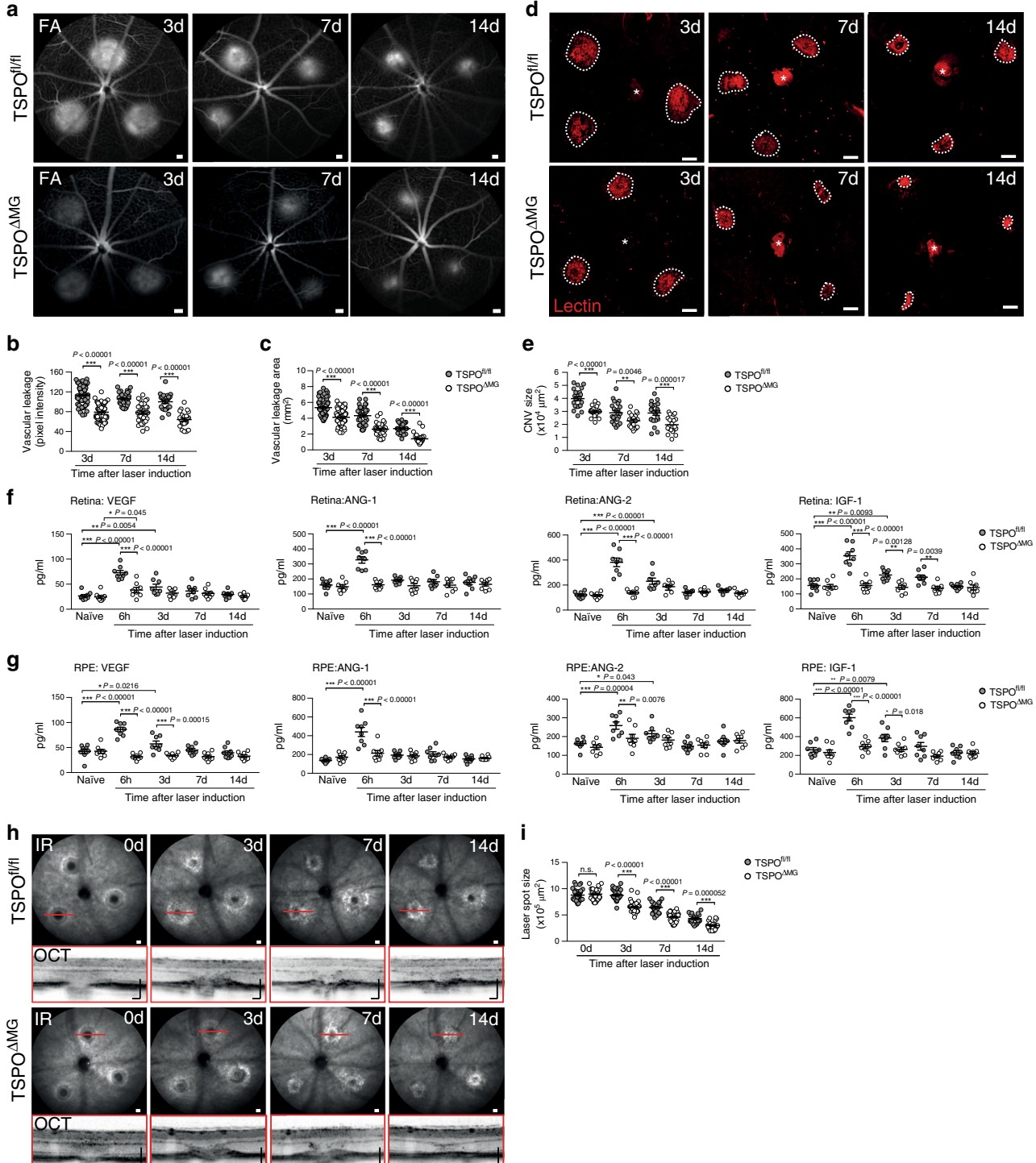

**Fig. 4 TSPO deficiency inhibits laser-induced vascular leakage and pathological CNV in mice. a** Representative late phase fundus fluorescein angiography (FFA) images at indicated time points post laser injury. Scale bar: 200 µm. **b** Quantification of vascular leakage intensity after laser-induced CNV. 3 d $n = 85$; 7 d $n = 36$; and 14 d $n = 22$ retinas from individual mice. FA fluorescein angiography. **c** Quantification of vascular leakage area after laser-induced CNV. DMSO/XBD173 3 d $n = 91/79$; 7 d $n = 42/33$; 14 d $n = 30$ retinas from individual mice. **d** Representative laser-induced CNV stained with isolectin B4 in RPE/choroidal flat mounts. Scale bar: 100 µm; **e** Quantification of laser-induced CNV area in RPE/choroidal flat mounts. $n = 22$ RPE/choroids from individual mice. **f** Pro-angiogenic growth factor levels in retinas of naïve and lasered TSPO$^{fl/fl}$ and TSPO$^{\Delta MG}$ mice at indicated time points. $n = 8$ retinas/RPEs from individual mice. **g** Pro-angiogenic growth factor levels in RPE/choroids of naïve and lasered TSPO$^{fl/fl}$ and TSPO$^{\Delta MG}$ mice at indicated time points. $n = 8$ retinas/RPEs from individual mice. **h** Representative infrared (IR) fundus images at indicated time points post laser injury. Lower panel shows OCT scan from one laser spot marked by a red line. Scale bar: 200 µm. **i** Quantification of laser spot size. 0 d $n = 45$, 3 d $n = 39$, 7 d $n = 33$, and 14 d $n = 25$ eyes from individual mice. Data show mean ± SEM and a linear mixed model was used for statistical analyses; *$P < 0.05$; **$P < 0.01$; and ***$P \leq 0.001$. n.s., not significant. Source data are provided as a Source Data file.

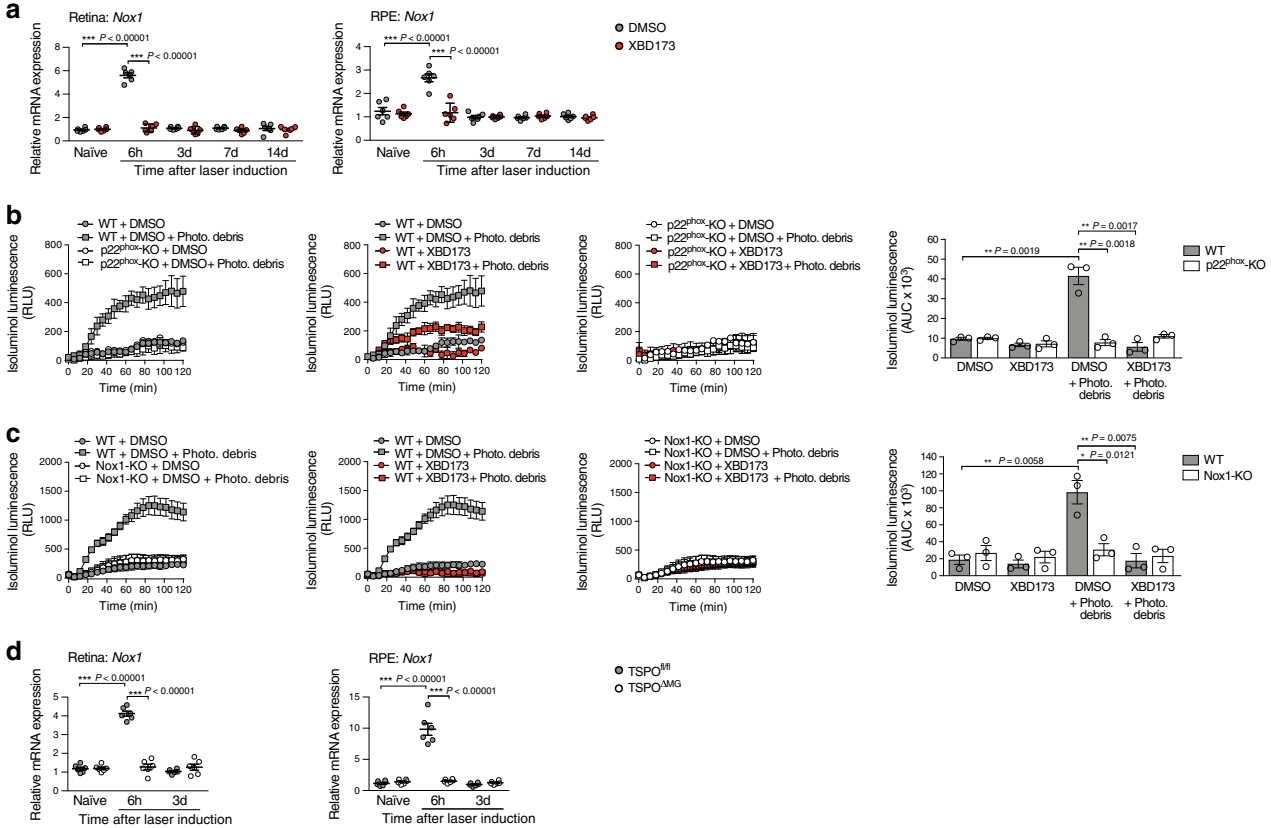

**Fig. 5 ROS production by primary microglia involves TSPO-dependent NOX1 activation. a** Laser-induced gene expression of *Nox1* in retina and RPE/choroid of DMSO-treated or XBD173-treated mice. Transcript levels for each enzyme were normalized to β-Actin. *n* = 6 retinas/RPEs from individual mice. **b** and **c** Quantification of extracellular ROS production by primary microglia from WT and p22phox-KO mice **b**, Nox1-KO mice **c**. Kinetics of ROS production and the area under the curve (AUC) are shown. Where indicated, primary microglia were stimulated with photoreceptor cell debris. *n* = 3 independent experiments. **d** Laser-induced gene expression of *Nox*1 in retina and RPE/choroid of TSPO^fl/fl and TSPO^ΔMG mice. Transcript levels for each enzyme was normalized to β-Actin. *n* = 6 retinas/RPEs from individual mice. Data shown as mean ± SEM. ROS data were analyzed using two-tailed unpaired Student's *t* test. A linear mixed model was used for laser-CNV data; *P < 0.05; **P < 0.01; and ***P ≤ 0.001. Source data are provided as a Source Data file.

lower *Cd68* expression in the retina after laser injury (Supplementary Fig. 1e, f). In addition, the lesion-associated formation of HMW1 TSPO was decreased in retinas from Nox1-KO mice compared to wildtype littermates (Fig. 7d, e). While IL-6 secretion did not change in Nox1-KO retinas, CCL2 levels were significantly reduced compared to WT retinas (Fig. 7f). Similar results were observed in RPE/choroidal flat mount analyses, where NOX1 deficiency reduced Iba1+ cell infiltration (Fig. 7g–i) and disease-associated expression of *Cd68* and *Tspo* (Supplementary Fig. 1e, f). Furthermore, the formation of HMW1 and HMW2 TSPO was reduced (Fig. 7j, k) and CCL2 levels were decreased in the RPE/choroid from Nox1-KO mice, while IL-6 and IL-1β levels showed no differences (Fig. 7l).

Finally, we investigated the paracrine potential of extracellular ROS as damaging neurotoxins. Therefore, 661W photoreceptor cells were co-cultured with primary microglia stimulated with photoreceptor cell debris to induce their extracellular ROS production. While only few propidium iodide positive (PI+) photoreceptor cells could be detected via flow cytometry during co-culture with resting TSPO^fl/fl microglia (Supplementary Fig. 13), cell death of photoreceptor cells tremendously increased after induction of extracellular ROS production via photoreceptor cell debris stimulation of microglia. Strikingly, scavenging of extracellular ROS with L-ascorbic acid (vitamin C) or of global ROS with N-acetyl-cysteine (NAC) in the microglial medium strongly reduced cell death of co-cultured photoreceptor cells. Accordingly, a strong reduction of PI+ photoreceptor cells could

be observed when co-cultured with XBD173-treated microglia, microglia deficient for TSPO or microglia deficient for NOX1, which all were incapable of producing extracellular ROS after stimulation (Figs. 1m, 3m, and 5c). These findings clearly indicate a paracrine damaging effect of microglia-derived extracellular ROS on photoreceptors, confirming their in vivo potential as neurotoxins.

**Nox1-KO prevents laser-induced vascular leakage and CNV.** Finally, we next investigated the effect of NOX1 deficiency on laser-induced vascular leakage and CNV. Nox1-KO mice showed a strongly reduced vascular leakage (Fig. 8a–c) and CNV size (Fig. 8d, e). NOX1 deficiency did not affect laser-induced secretion of VEGF-A, ANG-1, ANG-2, and IGF-1 (Fig. 8f, g) but diminished lesion sizes and reduced fibrosis compared to WT mice (Fig. 8h, i). Thus, NOX1 is a critical modifier of disease progression and outcome in the laser-CNV model.

## Discussion
Chronic inflammation is a hallmark of many neurodegenerative disease and targeting of TSPO, a biomarker of gliosis, improved disease outcome in various preclinical model systems[15,16,34]. However, the exact molecular function of TSPO has not been elucidated in these studies. Here, we demonstrate a critical function of TSPO signaling in phagocyte-triggered neoangiogenesis of the retina, a model system that recapitulates key

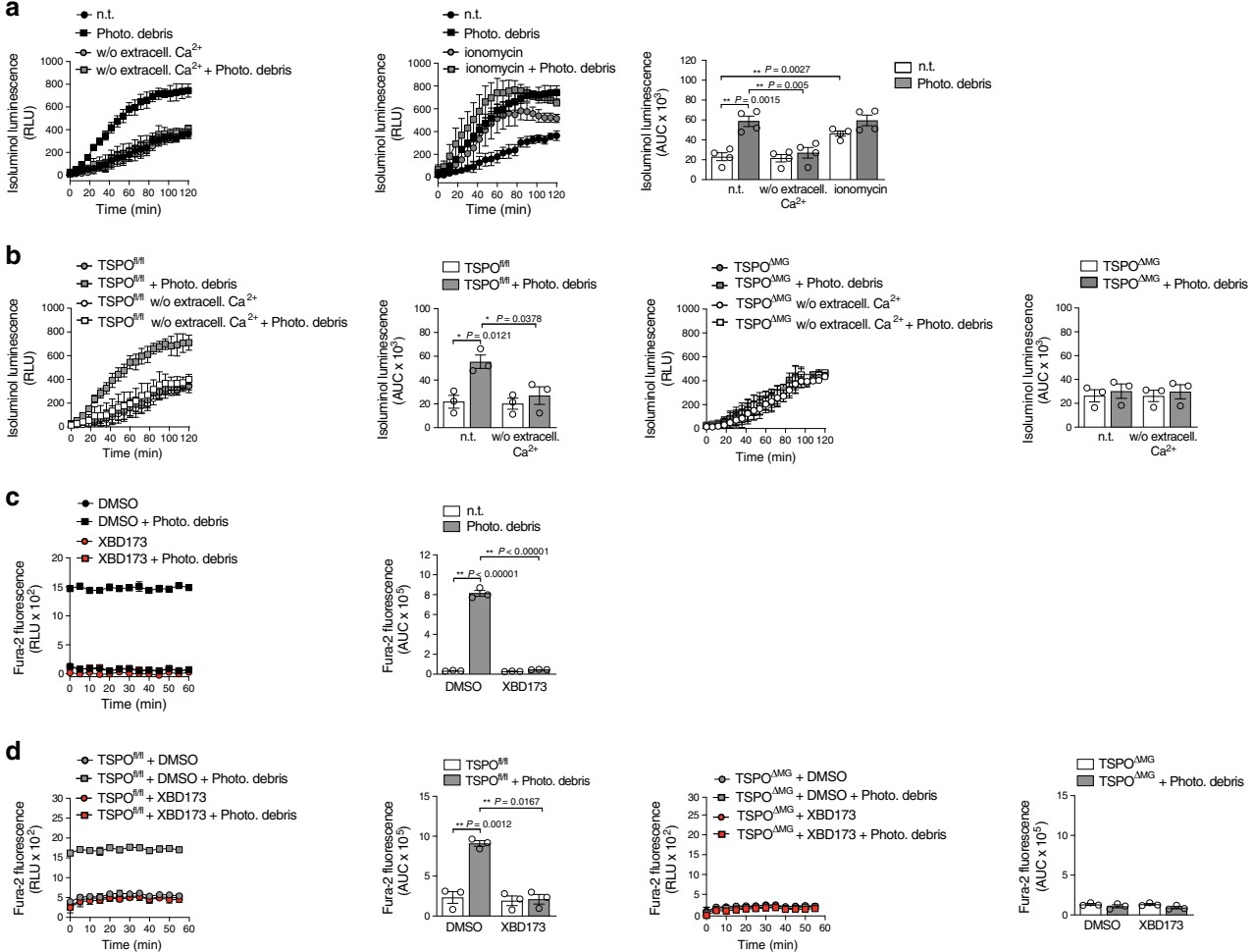

**Fig. 6 TSPO associated increase in cytosolic calcium is essential for Nox1-derived extracellular ROS production. a** and **b** Quantification of extracellular ROS production by primary microglia from WT **a**, TSPO$^{fl/fl}$, and TSPO$^{\Delta MG}$ mice **b**. Kinetics of ROS production and the area under the curve (AUC) are shown. Where indicated, primary microglia were stimulated with photoreceptor cell debris and cytosolic Ca$^{2+}$ was increased with the Ca$^{2+}$ ionophore ionomycin as a positive control. $n = 4$ **a**, 3 **b** independent experiments. **c** and **d** Quantification of cytosolic calcium levels in primary microglia from WT **c**, TSPO$^{fl/fl}$, and TSPO$^{\Delta MG}$ mice **d**. Where indicated, primary microglia were stimulated with photoreceptor cell debris. $n = 3$ independent experiments. Data show mean ± SEM; two-tailed unpaired Student's $t$ test, *$P < 0.05$, **$P < 0.01$. n.t. non-treated. Source data are provided as a Source Data file.

pathological features of neovascular AMD. As summarized in a schematic model (Fig. 9), we postulate that TSPO is critical for the Ca$^{2+}$ associated, NOX1-mediated production of extracellular ROS in retinal phagocytes. Targeting TSPO by gene knockout or by using the specific ligand XBD173 limits retinal innate immune cell responses and pathological angiogenesis (Fig. 9).

A previous study reported neuroprotective effects of the TSPO ligand XBD173 in retinal ischemia and the effect was mainly confined to retinal Müller cells with less effect on microglia[35]. We have demonstrated before that TSPO is constitutively expressed in the RPE and showed no inflammation-induced expression[14], suggesting that the retinal TSPO increase after laser-injury mainly stems from resident and invading mononuclear phagocytes. Our Western blots revealed the presence of HMW TSPO in the retina and RPE after laser injury in vivo. The appearance of these HMW bands could be due to post-translational modification and subsequent oligomerization, since a putative phosphorylation motif has been identified in the C-terminal domain of TSPO[36]. Also, a study using colonic cells showed that the TSPO ligand PK11195 can induce TSPO polymerization by stabilizing the dimeric form[37]. However, our data showed that XBD173 prevented the formation of these HMW bands. Therefore, the precise composition and role of these TSPO proteins in the retina deserves

further studies. Concomitant with reducing TSPO expression, XBD173 treatment also reduced the secretion of pro-inflammatory cytokines after laser injury. This is in line with a previous mouse study on parkinsonism, reporting that XBD173-specific transcriptional changes includes pathways related to cytokine production[38]. Our study also showed that XBD173 reduced pro-angiogenic factor expression and subsequently diminished CNV via modulation of phagocytes. Notably, while *Vegf, Ang1,* and *Ang 2* were expressed by phagocytes in vivo, only VEGF was produced and secreted by stimulated microglia. Indeed, Iba1$^+$ mononuclear phagocytes actively produced VEGF during laser injury[39] and macrophage depletion correlates with reduced VEGF expression and laser-induced CNV[40], while ANG1 and ANG2 are not produced by microglia. Furthermore, autocrine IL-1β can potently induce VEGF production by RPE cells[41] and the reduced IL-1β levels found upon XBD173 treatment may also dampen RPE-derived VEGF levels indicating a complex paracrine interplay of pro-inflammatory and pro-angiogenic factors on the tissue and cellular level.

To determine if TSPO function in mononuclear phagocytes modulates their reactivity during laser-CNV, we generated conditional TSPO-KO mice. Our data demonstrated that microglia lacking TSPO showed no morphological differences compared to

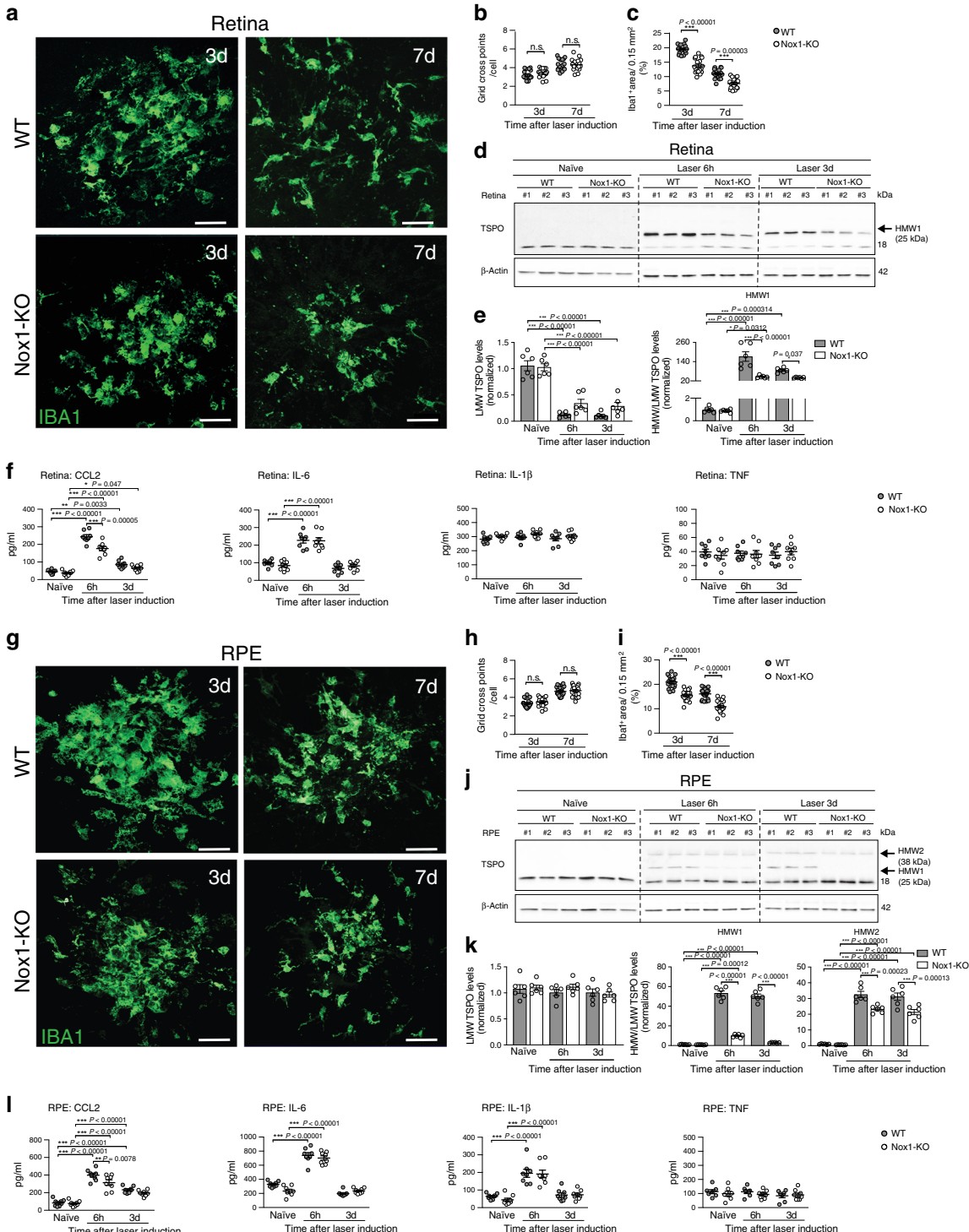

**Fig. 7 NOX1 deficiency reduces mononuclear phagocyte reactivity in laser-induced CNV in mice. a** Representative images show accumulation of Iba1+ cells within the laser lesion in retinal flat mounts. Scale bar: 50 μm. **b** Quantification of Iba1+ cell morphology within laser lesions. 3 d $n$ = 18; 7 d $n$ = 22 spots. **c** Quantification of Iba1+ area of the laser lesions. 3 d $n$ = 18; 7 d $n$ = 22 spots. **d** TSPO protein levels in retinal cell extracts of naïve and lasered WT and Nox1-KO mice at indicated time points. Each lane represents an individual retina. Dotted line indicates individual blots, which were processed in parallel. **e** Densitometric analysis of western blots. LMW TSPO signals were normalized to β-Actin and HMW:LMW TSPO ratio determined. $n$ = 6 retinas from two independent experiments. **f** Cytokine levels in retinas of naïve and lasered WT and Nox1-KO mice at indicated time points. $n$ = 8 retinas from individual mice. **g** Representative images of Iba1+ cells within the laser lesion in RPE/choroidal flat mounts. Scale bar: 50 μm. **h** Quantification of Iba1+ cell morphology within laser lesions. 3 d $n$ = 18, 7 d $n$ = 22 spots. **i** Quantification of Iba1+ area of the laser lesions. 3 d $n$ = 18, 7 d $n$ = 22 spots. **j** Western blots showing TSPO expression in RPE/choroidal cell extracts of naïve and lasered WT and Nox1-KO mice at indicated time points. **k** Densitometric analysis of western blots. $n$ = 6 RPE/choroids from two independent experiments. LMW lower molecular weight; HMW higher molecular weight. **l** Pro-inflammatory cytokine levels in RPE/choroids of naïve and lasered WT and Nox1-KO mice. $n$ = 8 RPE/choroids from individual mice. Data show mean ± SEM. A linear mixed model was used for statistical analyses, *$P$ < 0.05; **$P$ < 0.01; and ***$P$ ≤ 0.001. Source data are provided as a Source Data file.

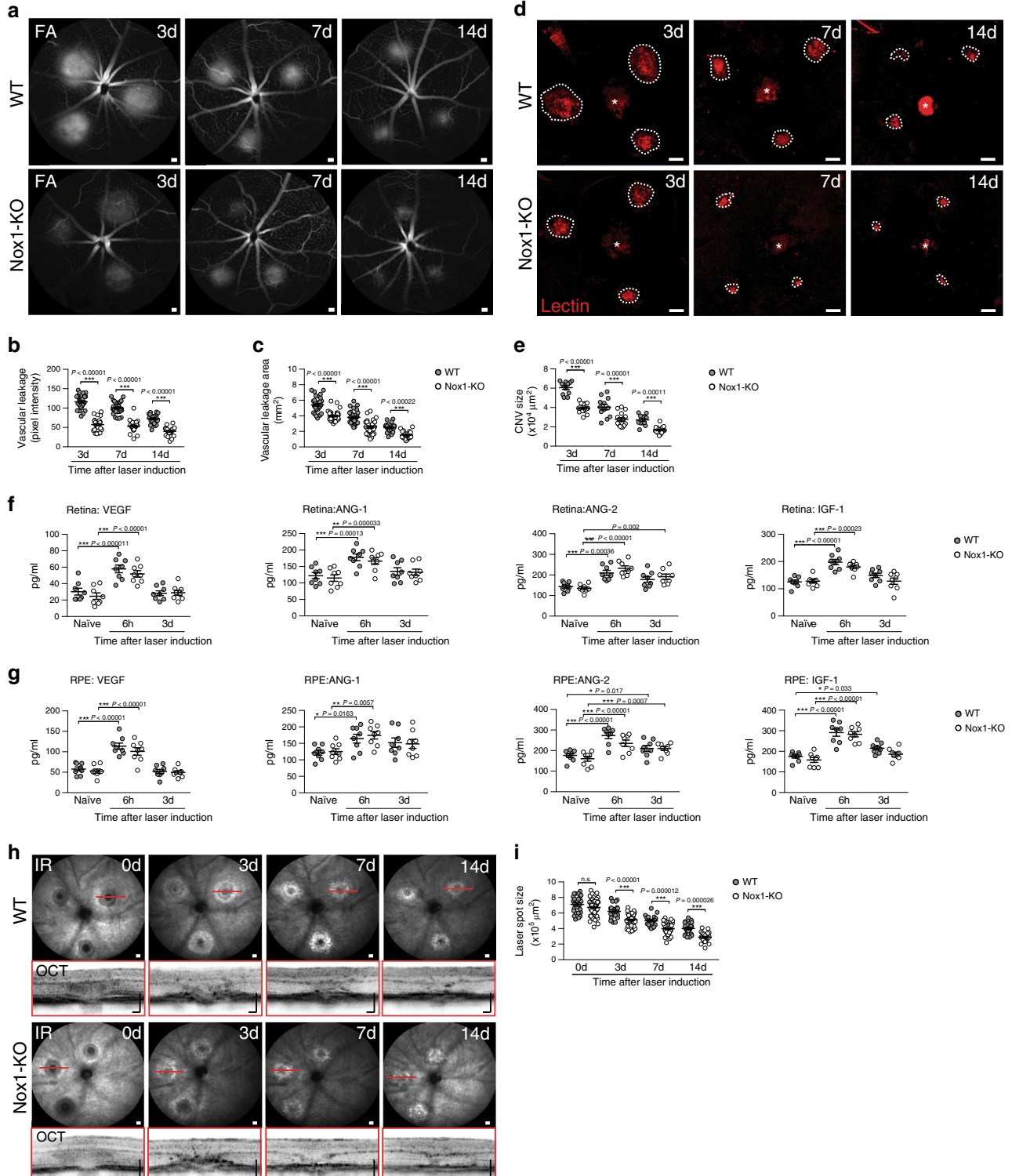

**Fig. 8 NOX1 deficiency limits laser-induced vascular leakage and pathological CNV in mice. a** Representative late phase fundus fluorescein angiography (FFA) images at indicated time points post laser injury. Scale bar: 200 μm. **b** Quantification of vascular leakage intensity after laser-induced CNV. 3 d/7 d n = 32 and 14 d n = 22 eyes from individual mice. **c** Quantification of vascular leakage area after laser-induced CNV. 3 d/7 d n = 32 and 14 d n = 22 eyes from individual mice. **d** Representative laser-induced CNV stained with isolectin B4 in RPE/choroidal flat mounts. Scale bar: 100 μm; FA fluorescein angiography. **e** Quantification of laser-induced CNV area in RPE/choroidal flat mounts. WT/Nox1-KO n = 12/18 RPE/choroids from individual mice. **f** Pro-angiogenic growth factor levels in retinas of naïve and lasered WT and Nox1-KO mice at indicated time points. n = 8 eyes from individual mice. **g** Pro-angiogenic growth factor levels in RPE/choroids of naïve and lasered WT and Nox1-KO mice at indicated time points. n = 8 eyes from individual mice. **h** Representative infrared (IR) fundus images at indicated time points post laser injury. Lower panel shows OCT scan from one laser spot marked by a red line. Scale bar: 200 μm. **I** Quantification of laser spot size. WT/Nox1-KO 0 d n = 28/68, 3 d n = 22/51, 7 d n = 22/45,14 d n = 22/25, n = 25 eyes from individual mice. Data show mean ± SEM. A linear mixed model was used for statistical analyses, *P < 0.05; **P < 0.01; and ***P ≤ 0.001. n.s., not significant. Source data are provided as a Source Data file.

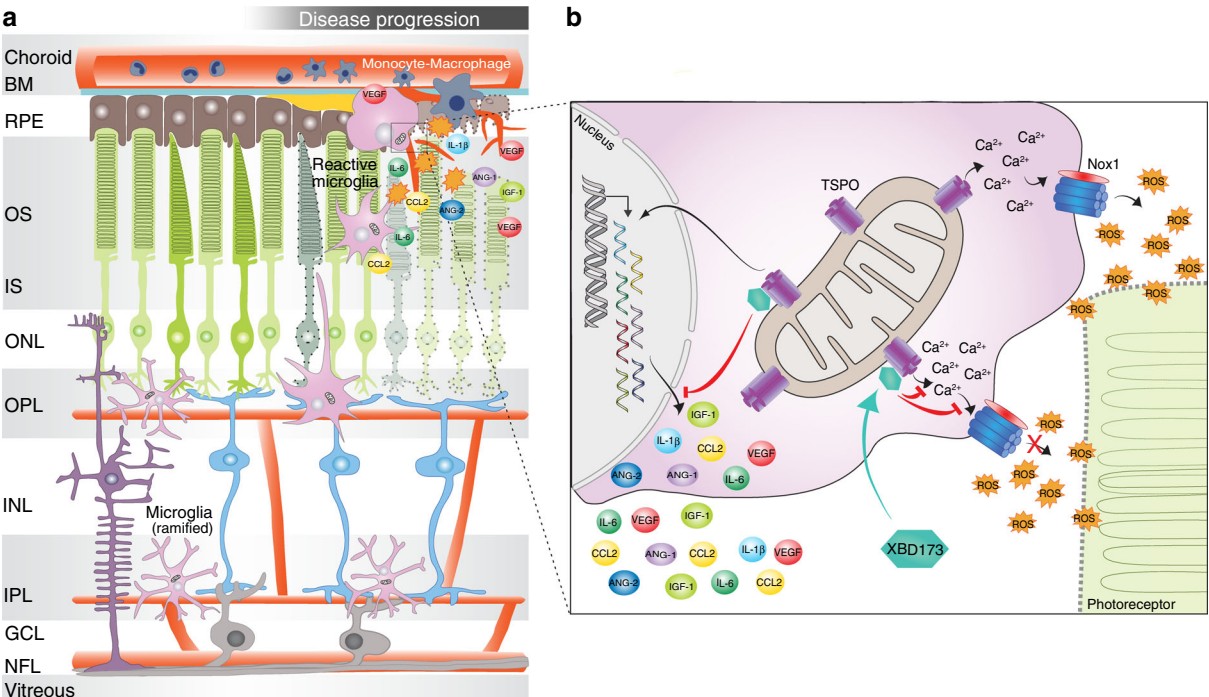

**Fig. 9 Model of TSPO-mediated ROS production in retinal phagocytes. a** In the healthy retina, resident microglia populate the plexiform layers. With their long protrusions, they constantly scan their environment and phagocytose cell debris. Different insults in the RPE and photoreceptor layer rapidly alert microglia. Resident microglia transform into ameboid phagocytes, migrate to the lesion sites and recruit macrophages from the periphery. **b** In response to these pathological signals, microglia increase pro-inflammatory and pro-angiogenic cytokine expression to resolve neuroinflammation and promote tissue recovery. Reactive microglia also upregulate mitochondrial TSPO leading to increased cytosolic calcium levels, which is essential for NOX1-mediated ROS production. Chronic activation of microglia may be detrimental and promote retinal degeneration. Binding of the synthetic ligand XBD173 to TSPO limits inflammatory responses and inhibits the increase of cytosolic calcium levels thus preventing from ROS damage. XBD173 supports the conversion of reactive microglia towards a neuroprotective phenotype, limiting pathological CNV. BM Bruch's membrane; OS outer segment; IS inner segment; ONL outer nuclear layer; OPL outer plexiform layer; INL inner nuclear layer; IPL inner plexiform layer; GCL ganglion cell layer; NFL nerve fiber layer.

control cells. The microglia from TSPO$^{\Delta MG}$ mice showed no alterations in predictors of mitochondrial health, including mitochondrial morphology or MMP. Analysis of total cellular ATP levels showed that microglia use both glycolysis and mitochondrial respiration to generate ATP and that this was not changed in TSPO knockout microglia. This is in contrast to studies reporting reduced mitochondrial metabolism in mouse and human microglia cell lines after TSPO knockdown or knockout[42,43]. This discrepancy could be due to the fact that endogenous levels from immortalized cultured cell lines and primary microglia differ in their TSPO function. Thus, a report on liver-specific TSPO-KO mice also observed neither mitochondrial ultrastructural alterations nor membrane potential or ATP level differences[22].

We showed here that conditional deletion of TSPO in long-lived retinal phagocytes was phenocopying the beneficial effect of XBD173 treatment on laser-CNV. In line with this, two recent studies demonstrated that astroglia-specific TSPO-KO was protective in a mouse model for MS[44] and that a cardiac-specific TSPO-KO protected from pressure overload induced heart failure[45]. Involvement of TSPO in ROS production was suggested before[20,46,47] and chronic ROS production can be a driving force for disease progression[48]. The regulated production of ROS is mediated by members of the NOX enzyme family[49]. NOX2 is the predominant source of ROS in phagocytes[50], but several studies also described a role of Nox1-dependent and Nox4-dependent ROS production in microglia from different neurological diseases[51–53]. While only a few in vivo studies investigated the role of NOX2 or NOX4 in deficient mice[53–56], most of the research on microglia was performed with cell lines. NOX enzymes were

either knocked down[46,57,58] or ROS production was chemically inhibited with diphenyleneiodonium or apocyanin[46,51,53,58]. While commonly termed specific NOX inhibitors, these compounds show numerous side effects and their specificity is questioned[59]. Animal studies with genetically modified NOX enzymes in eye diseases[56,60] are scarce. By using different Nox-KO mouse strains, we showed that microglia produce extracellular ROS exclusively via NOX1, while other NOX enzymes were dispensable for this response. Furthermore, TSPO was crucial for the induction of NOX1-dervied ROS. TSPO was implicated in mitochondrial ROS production before, which was associated with cellular signaling functions[20]. However, we did not observe ROS production in the cytosol or the mitochondrial matrix after stimulation. Notably, we could show that NOX1-derived extracellular ROS induced photoreceptor cell death in a paracrine manner, confirming their potential as damaging neurotoxins. Importantly, *Nox1*, but not other *Nox* or *Duox* enzymes were up-regulated in vivo after laser-injury and accordingly, NOX1-deficiency improved disease outcome in these mice, while other features of microglia reactivity were not affected.

We propose that TSPO acts as a regulatory node and regulates microglia functions through both NOX1-dependent and NOX1-independent mechanisms. NOX1 is terminally in the TSPO–NOX1 axis, and we showed that TSPO regulates NOX1-mediated production of ROS that can kill photoreceptor cells in a paracrine manner. However, TSPO-mediated regulation of the production of pro-inflammatory cytokines and angiogenic growth factors, is clearly independent from NOX1. The TSPO-regulated NOX1-dependent mechanism is crucial in the laser-induced CNV model, as Nox1-KO mice show the same beneficial effects on

CNV and wound healing as XBD173 treatment or microglia-specific TSPO-KO.

A TSPO-dependent regulation of NOX enzymes was described before[46,47] and heme and/or cholesterol transport was suggested as a possible regulatory mechanism[47]. Furthermore, two other studies demonstrated an interaction of TSPO with the channel VDAC1 and a potential role in redox homeostasis via $Ca^{2+}$ [20,46]. Therefore, we decided to analyze the role of TSPO in calcium homeostasis and subsequent NOX1-derived ROS production. In accordance with a former study demonstrating elevated cytosolic $Ca^{2+}$ levels in TSPO-overexpressing cells[46], we observed a strong reduction in cytosolic $Ca^{2+}$ in TSPO-deficient microglia. Notably, the TSPO-mediated increase of cytosolic $Ca^{2+}$ levels was prevented by XBD173, which subsequently abolished not only Nox1 expression, but also acute NOX1-derived ROS production after stimulation. Interestingly, in the absence of extracellular $Ca^{2+}$, Nox1 expression and NOX1-dependent ROS production upon stimulation was significantly reduced in primary microglia, indicating that extracellular ROS production is regulated via both, NOX1 protein expression and direct activation in a $Ca^{2+}$-dependent manner. While DUOX1-2 and NOX5 are directly activated via their EF-hand calcium-binding domains[49], ROS production in microglia depends on the catalytic subunit p22[phox]. This suggests that the rise in cytosolic $Ca^{2+}$ indirectly activates NOX1 via calcium-dependent signaling mechanisms in the cytosol. The identification of these signaling cascades in microglia will be a topic for further investigations.

## Methods

**Animals**. Mice were housed under specific pathogen–free conditions in individually ventilated caging (IVC) systems (GM 500, Tecniplast® Greenline) with a maximum cage density of five adult mice per cage. Light was adjusted to a 12 h/12 h light/dark cycle with light on at 6 a.m., temperature and relative humidity were regulated to 22 ± 2 °C and 45-65% relative humidity. Mice were fed irradiated phytoestrogen-free standard diet for rodents (Altromin 1314; 59% carbohydrates, 27% protein, 14% fat) and had access to food and acidified water ad libitum. Cx3cr1[CreERT2]:Tspo[fl/fl] mice (TSPO[ΔMG]), which were obtained by breeding Cx3cr1[CreERT2] mice[28] and TSPO[fl/fl] mice[22], purchased from the Jackson Laboratory (Bar Harbor, ME, USA). Nox1-KO[61], Nox2-KO[29], Nox4-KO[62], and p22[phox]-KO[63] mice are on C57BL/6J background. C57BL/6J mice and homozygous transgenic knock-out mice and corresponding wild type littermates, 8–10 weeks old, were used for experiments. All experimental procedures complied with the German law on animal protection and the ARVO Statement for the Use of Animals in Ophthalmic and Vision Research. The animal protocols used in this study were reviewed and approved by the governmental body responsible for animal welfare in the state of Nordrhein-Westfalen (Landesamt für Natur, Umwelt und Verbraucherschutz Nordrhein-Westfalen, Germany; approval no. 84-02.04.2017.A034).

**XBD173 and Tamoxifen administration**. The phenylpurine XBD173 (AC-5216, emapunil) was obtained by custom synthesis from APAC Pharmaceuticals. The mice received intraperitoneal injections of XBD173 at a dose of 10 mg/kg dissolved in DMSO or DMSO as a vehicle control daily starting one day before laser photocoagulation. Tamoxifen powder (T5648, Sigma-Aldrich) was partially dissolved in 100% ethanol and vortexed for 5 min. Filter-sterilized corn oil (C8267, Sigma-Aldrich) was added to a 9:1 oil:ethanol mixture ratio to a final concentration of 20 mg/ml tamoxifen and incubated at 37 °C until full dissolution. The prepared tamoxifen working solution was stored at −20 °C protected from light. For induction of Cre recombinase activity, 4−6-week-old TSPO[ΔMG] mice and littermates carrying the respective loxP-flanked alleles but lacking expression of Cre recombinase (TSPO[fl/fl]), were treated 4 weeks before start of experiments with 4 mg tamoxifen injected intraperitoneally twice 2 days apart.

**Laser photocoagulation**. For laser photocoagulation mice were anesthetized with a mixture of ketamine (100 mg/kg body weight, Ketavet; Pfizer Animal Health) and xylazine (5 mg/kg body weight, 2% Rompun; Bayer HealthCare) diluted in 0.9% sodium chloride by intraperitoneal (i.p.) injection and their pupils dilated with a topical drop of phenylephrine 2.5%–tropicamide 0.5%. A slit-lamp-mounted diode laser system (Quantel Medical Vitra, 532 nm green laser, power 100 mW, duration 100 ms, and spot size 100 μm) was used to generate three equal laser burns around the optic nerve in each eye with a cover glass as a contact lens[39]. For gene expression and protein analysis, 20 laser burns were applied to both eyes. To validate rupture of Bruch's membrane, infrared images were recorded using

Spectralis™ HRA/OCT device (Heidelberg Engineering) to analyze post-laser retinal structure and laser lesion size in vivo. In case of cataract and corneal epithelial edema before laser photocoagulation, unsuccessful laser burn without Bruch's membrane rupture, or severe choroidal hemorrhages, eyes were excluded from further analysis.

**Fundus photography and FFA**. Vascular leakage was analyzed 3, 7, and 14 days after laser photocoagulation. After anesthesia and pupil dilatation, mice received intraperitoneal injection of 0.1 ml of 2.5% fluorescein (Alcon®) diluted in 0.9% sodium chloride. Late phase angiograms were recorded 10 min after fluorescein injection using Spectralis™ HRA/OCT (Heidelberg Engineering). Simultaneously, infrared fundus images (IR) were acquired to analyze the laser lesion size. The size of laser lesions and vascular leakage was determined using the measuring tool of the HEYEX software (Heidelberg Engineering). The analysis of vascular leakage by measuring pixel intensities was performed as described previously[24]. In brief, pixel intensity was quantified in two regions of interest (ROI) within and one ROI outside each laser lesion using ImageJ. The background pixel intensity was then subtracted from the laser lesion values and the data of three laser lesions averaged to obtain the mean laser-induced leakage per eye.

**Flat mounts, immunohistochemistry, and image analysis**. Mice were euthanized by cervical dislocation and the eyes enucleated and fixed in 10% neutral buffered formalin (NBF) for 2 h at room temperature (RT). The dissected retinal and RPE/choroidal flat mounts were permeabilized and blocked overnight in Perm/Block buffer (5% normal donkey serum (NDS), 0.2% BSA, 0.3% Triton X-100 in PBS) at 4 °C. The flat mounts were subsequently incubated with a polyclonal rabbit anti-IBA1 antibody (1:500 diluted in Perm/Block, 019-19741, Wako) for 48 h at 4 °C. After washing three times with PBST-X (0.3 % Triton X-100 in PBS), the flat mounts were incubated for 1 h with donkey anti-rabbit AlexaFluor[TM]488 (1:1000 diluted in Perm/Block, A21206, Invitrogen). RPE/choroidal flat mounts were stained in addition with TRITC-conjugated isolectin B4 from Bandeiraea simplicifolia (1:100 diluted in Perm/Block, L5264, Sigma-Aldrich). After several washing steps, retinal and RPE/choroidal flat mounts were mounted on a microscope slide and embedded with fluorescence mounting medium (Vectashield HardSet H-1400, Vector Labs). Images were taken with a Zeiss Imager.M2 equipped with an Apo-Tome.2. Morphological analysis of Iba1[+] mononuclear phagocytes in lasered retinas and RPE/choroidal flat mounts was done using a grid system[24] and the evaluation of Iba1[+] area per laser lesion was performed using ImageJ. The average grid crossing points and Iba1[+] area per laser lesion was calculated. Areas of CNV in RPE/choroidal flat mounts were measured with the spline function of the graphic tool included in the ZEN software (Zeiss). The average CNV area per eye was calculated. Morphometric parameters in retinal flat mounts were analyzed using MotiQ, a fully automated analysis software. MotiQ was developed as an ImageJ plugin in Java and is publicly available (https://github.com/hansenjn/MotiQ). All segmentation and quantification were performed on 2D mean intensity projections (MIPs) of 3D image data.

**RNAScope® ISH**. RNAScope® ISH (ACD, RNAscope® Multiplex Fluorescent Reagent Kit v2) was performed according to the manufacturer's instructions with some modifications. In brief, mice were euthanized by cervical dislocation and the eyes enucleated and fixed in 10% NBF for 2 h at RT. The dissected RPE/choroidal flat mounts were pre-treated with protease 3 for 20 min at RT and washed 3× with 1x Wash buffer for 10 min at RT. All hybridization, amplification, and HRP signal detection steps were done according to the protocol provided by ACD. The following probes were used in this study: Mm-Aif1-C3, ACD 319141; Mm-Angpt1-C1, ACD 449271; Mm-Angpt2-C1, ACD 406091; and Mm-Vegf-ver2-C1, ACD 412261. C1 probes were labeled with TSA® Plus Fluorophore Cyanine 3 and C3 probes with TSA® Plus Fluorophore Fluorescein.

**Quantitative PCR**. RNA was isolated from retinal and RPE/choroidal tissue or isolated primary microglia using the RNeasy Micro Kit (Qiagen) according to the manufacturer's instructions. First-strand complementary DNA (cDNA) was synthesized from the total mRNA using the RevertAid™ H Minus First strand cDNA Synthesis Kit (Thermo Scientific). Transcript levels of Cd68, Duox1, Duox2, Nox1, Nox2, Nox3, Nox4, and Tspo were analyzed by quantitative real-time PCR performed in LightCycler® 480 II (Roche) with either SYBR® Green (Takyon No Rox SYBR Master Mix dTTP blue, Eurogentec) or probe-based (LightCycler® 480 Probes Master, Roche) detection. Actin and Atp5b were used as housekeeping genes. Measurements were performed in technical duplicates and delta delta CT threshold calculation was used for relative quantification of results. UPL probes used for probe-based detection were purchased from Roche. Primers are as follows: Ang1, 5′-aggagcacgcagctagaca-3′ (forward), 5′-acccacgtccatgtcacag-3′ (reverse), probe #60; Ang2, 5′-ccaccagtggcatctacaca-3′ (forward), 5′-tttctccgctctgaacaagg-3′ (reverse), probe #48; Cd68, 5′-ctctctaaggctacaggctgct-3′ (forward), 5′-tcacggttg-caagagaaaca-3′ (reverse), probe #27; Nox1, 5′-ggatggatctctcgcttctg-3′ (forward), 5′-aatgctgcatacatcactgtca-3′ (reverse), probe #19; Tspo, 5′-actgtattcagccatggggta-3′ (forward), 5′-accatagcgtcctctgtgaaa-3′ (reverse), probe #33; Vegf, 5′-aaaaac-gaaagcgcaagaaa-3′ (forward), 5′-tttctccgctctgaacaagg-3′ (reverse), probe #1; Atp5b, 5′-ggcacaatgcaggaaagg-3′ (forward), 5′-tcagcaggcacatagatagcc-3′ (reverse), probe

#77; *Nox2*, 5′-ggttccagtcgcgtgttgct-3′ (forward), 5′-gcggtgtgcagtgctcatcat-3′ (reverse); *Nox3*, 5′-gtgataacaggcttaaagcagaaggc-3′ (forward), 5′-ccactttcccctacttgactt-3′ (reverse); *Nox4*, 5′-ggagactggacagaacgattc-3′ (forward), 5′-tgtataacttagggtaatttcta-gagtgaatga-3′ (reverse); *Duox1*, 5′-agcccctgaaagaaccctac-3′ (forward), 5′-tccccatgcgggatgtaaatg-3′ (reverse); *Duox2*, 5′-tccattagtgagtctgattgtc-3′ (forward), 5′-gtttgtcaaggacctgcagact-3′ (reverse); *Actin*, 5′-aggaggagcaatgatcttg-3′ (forward), 5′-agacctgtacgccaacacag-3′ (reverse).

**PCR of ΔTSPO**. The ΔTSPO PCR was performed with primers spanning the *loxP*-flanked exons 2 and 3 of the *Tspo* gene. PCR amplification of cDNA isolated from retina and RPE/choroidal tissues from TSPO$^{fl/fl}$ mice resulted in a 526 bp fragment and from TSPO$^{\Delta MG}$ mice in 526 and 176 bp fragment. Primers are as follows: *Tspo* P1, 5′-taccaacctctgtgcgcag-3′ (forward), *Tspo* P2, 5′-atgctctaagggcatgcctg-3′ (reverse).

**ELISA**. The concentration of cytokines in total retinal, RPE, or primary microglia lysates were measured by ELISA. Tissue samples were sonicated in PBS supplemented with protease and phosphatase inhibitors (Complete protease inhibitor cocktail, Roche). CCL2/JE/MCP-1 (DY479), IGF-1 (DY791), IL-1beta/IL-1F2 (DY401), IL-6 (DY406), TNF (DY410), VEGF (DY493) DuoSet®, and ANG-2 (MANG20) Quantikine® ELISA's were purchased from R&D Systems, ANG-1 (MBS727480) was purchased from MyBioSource.

**Western blot**. TSPO protein levels of were analyzed under non-reducing conditions. For this, tissue samples and primary microglia were lysed by sonication in ice cold PBS supplemented with protease and phosphatase inhibitors (Complete protease inhibitor cocktail, Roche). Protein concentration was determined by BCA Protein Assay according to the manufacturer's instructions (Thermo Scientific). Equal amounts of samples were heated in SDS sample buffer containing only 0.2% SDS and no β-mercaptoethanol and then loaded onto 12% tris–glycine poly-acrylamide gels and run under standard conditions. For immunoblotting, proteins were electrophoretically transferred onto a 0.45 μm nitrocellulose membrane (Bio Rad) at 100 V for 1 h. Membranes were blocked with 5% nonfat dried milk powder in TBS-T before incubation with primary antibody against TSPO (1:1000 dilution, rabbit polyclonal, ab109497, Abcam) or Actin (1:1000 dilution, chicken anti-b-Actin clone AC-15, A5441, Sigma-Aldrich). After several washing steps in TBS-T, membranes were incubated with horseradish peroxidase-conjugated secondary antibodies (1:4000 dilution, Dako) and the immune complex was visualized by a MultiImage II system (Alpha Innotech). PageRuler pre-stained protein ladder (Thermo Scientific) was used for identification of protein size. Band intensities were quantified using ImageJ.

**Isolation, culture, and stimulation of primary microglia**. Brains from 8 to 10-week-old mice were harvested in cold HBSS and after removal of the meninges enzymatically dissociated into single cell suspensions with a papain-based Neural Tissue Dissociation Kit (Miltenyi Biotec) on the GentleMACS® Dissociator (Miltenyi Biotec) according to the manufacturer's instructions. The single cell suspension was passed through a 70 μm cell strainer and depleted of myelin by suspension in 30% isotonic Percoll® (GE Healthcare Life Sciences) followed by a 10 min centrifugation at $700 \times g$ at 4 °C. Primary microglia were enriched by magnetic cell sorting using CD11b MicroBeads (Miltenyi Biotec) according to the manufacturer's instructions. Cells were seeded into well plates in Dulbecco's modified Eagle's medium (DMEM) supplemented with 10% fetal calf serum (FCS) and pre-treated with 50 μM XBD173 (APAC Pharmaceuticals), 50 μM Ro5-4864, 50 μM Etifoxine, 50 μM PK11195 (all purchased from Sigma-Aldrich), 50 μM FGIN-1-27 (Tocris), or DMSO as vehicle control for 1 h at 37 °C as indicated. Primary microglia were stimulated with 661W photoreceptor cell debris by synchronization at $850 \times g$ for 5 min at 4 °C, with ultrapure LPS from *E. coli* O111:B4 (100 ng/ml) (Invitrogen) or PMA (1 ng/ml) (Sigma-Aldrich). After three washing steps, cells were incubated in Hanks' balanced salt solution (HBSS) with Ca$^{2+}$ and Mg$^{2+}$ supplemented with 5% heat-inactivated normal mouse serum (Dunn Lab). Where indicated, carbonyl cyanide 3-chlorophenylhydrazone (CCCP) (200 μM), ionomycin (100 μM), rotenone (100 μM) (all purchased from Sigma-Aldrich), was added to the medium after stimulation with 661W photoreceptor cell debris.

**Culturing of 661W photoreceptor cells**. 661W photoreceptor cells were a gift from Prof. Muayyad Al-Ubaidi (Department of Cell Biology, University of Oklahoma Health Sciences Center, USA). Cells were grown in a monolayer in DMEM high glucose supplemented with 10% heat-inactivated FCS and 1% penicillin–streptomycin (P/S) and maintained at 37 °C in a humidified atmosphere of 5% CO$_2$. At about 90% confluency, 661W cells were washed twice with 1x PBS and incubated with 1x trypsin-EDTA for 3 min at 37 °C to detach the adherent cells from the culture surface. The reaction was stopped by addition of an equal volume of serum-containing DMEM high glucose medium. Cells were collected, centrifuged at $300 \times g$ for 5 min and the cell pellet resuspended in DMEM high glucose supplemented with 10% heat-inactivated FCS and 1% P/S and counted using Trypan Blue exclusion in a Neubauer chamber.

**Co-culture of 661W cells and primary microglia**. Primary microglia at the bottom and 661W cells on top were separated in a trans-well culture with 0.4 μm inserts. As a basal culture medium, DMEM supplemented with 10% FCS, 1% Pen/Strep, was used. Primary microglia were seeded at a density of $2.5 \times 10^5$ cells/well in a 24-well plate and 661W cells at a density of $2.5 \times 10^4$ cells/well in 0.4 μm trans-well inserts. After 4 h, trans-wells with 661W cells were removed, placed in a new sterile 24-well plate and primary microglia were stimulated with 661W photo-receptor cell debris by synchronization at $450 \times g$ for 5 min at 4 °C. Where indicated, primary microglia were treated with 1 mM L-ascorbic acid (Sigma-Aldrich), 50 μM XBD173 or 50 mM NAC (Sigma-Aldrich). After incubation for 15 min at 37 °C, trans-wells with 661W cells were transferred back to the primary microglia and the trans-well co-culture was incubated for another 24 h.

**Flow cytometry**. Medium of 661W cells from the inlays was collected, cells were washed with 1x PBS and harvested using 1x Trypsin/EDTA for 20 s at RT. Collected medium and cells were centrifuged at $650 \times g$ at 4 °C for 5 min and resuspended in MACS buffer (PBS, 2 mM EDTA, 0.5% BSA). To assess the cell death of 661W cells after co-culturing with stimulated or unstimulated primary microglia propidium iodide (PI) staining (BD Pharmingen) was performed according to the manufacturer's instructions. Cells were analyzed via flow cytometry with a BD FACSCanto$^{TM}$ Flow Cytometer (BD Biosciences) and data were obtained and analyzed with BD FACSDiva$^{TM}$ software (BD Biosciences).

**Quantification of ROS production**. Microglia were seeded at a density of $1 \times 10^5$ cells/well in triplicates in DMEM + FCS in white or black 96-well plates for luminescence or fluorescence measurements, respectively. To analyze extracellular ROS production, primary microglia were incubated in 50 μM isoluminol (Sigma-Aldrich) and horseradish peroxidase (3.2 U/ml) (Merck Millipore) in HBSS after stimulation with 661W photoreceptor cell debris[27]. To analyze cytosolic ROS production, primary microglia were incubated in 20 μM DCF (Thermo Fisher Scientific) in HBSS for 15 min at 37 °C before stimulation[27]. To analyze ROS production into the mitochondrial matrix, primary microglia were incubated in 5 μM MitoSOX Red (Thermo Fisher Scientific) in HBSS for 15 min at 37 °C before stimulation. After stimulation with 661W photoreceptor cell debris, plates were transferred on ice to the respective plate reader pre-heated to 37 °C. Isoluminol luminescence or DCF fluorescence was measured at 1-min intervals using a TriStar$^2$ LB 942 Multimode Plate Reader (Berthold Technologies), and MitoSOX Red fluorescence was measured using a Tecan Infinite M 1000 microplate reader (Tecan Group).

**Quantification of calcium levels**. Primary microglia were seeded at a density of $1 \times 10^5$ cells/well in triplicates in DMEM + FCS in black 96-well plates. To analyze mitochondrial or cytosolic calcium levels, primary microglia were incubated in 2 μM Rhod2-AM (Enzo) or 2 μM Fura2-AM (Sigma-Aldrich) in HBSS for 15 min at 37 °C before stimulation with 661W photoreceptor cell debris, respectively. Rhod2-AM and Fura2-AM fluorescence was measured at 1-min intervals using a Tecan Infinite M 1000 microplate reader (Tecan Group).

**Mitochondrial staining of primary microglia**. For fluorescence microscopic analysis of the mitochondrial network of primary microglia, mitochondria were stained with the MMO-sensitive dye MitoTracker Red CMXRos (Thermo Fisher Scientific) at 100 nM for 15 min at 37 °C. Where indicated, primary microglia were treated with 200 μM CCCP as a positive control for mitochondrial network fragmentation.

**Analysis of MMP**. Primary microglia were incubated in 1 μM tetramethylrhodamine, ethyl ester (TMRE) (Abcam) in HBSS with Ca$^{2+}$ and Mg$^{2+}$ for 20 min at 37 °C and then stimulated with 661W photoreceptor cell debris or treated with 200 μM CCCP as a positive control. TMRE fluorescence was measured at 60-min intervals using a Tecan Infinite M 1000 microplate reader (Tecan Group).

**Analysis of cellular ATP levels**. Primary microglia were stimulated with 661W photoreceptor cell debris or treated with 200 μM CCCP as a positive control. Where indicated 2-DG (500 μM) or oligomycin A (10 μM) (all purchased from Sigma-Aldrich), was added to the medium after stimulation with 661W photo-receptor cell debris. ATP levels were determined using the CellTiter-Glo® Luminescent Cell Viability Assay (Promega) in accordance with the manufacturer's instructions. Chemiluminescence was measured at 60 min intervals using a TriStar$^2$ LB 942 Multimode Plate Reader (Berthold Technologies).

**Statistical analysis**. Statistical analysis was conducted on data from at least three independent experiments. Western blots from XBD173-treated, microglia-specific TSPO and Nox1-KO retinas or RPE/choroids were performed two times with three different biological samples each time. All micrographs shown are representative images of at least three independent experiments. Statistical analysis was performed using GraphPad Prism 8 software (v8.4.1, GraphPad software Inc.). Differences between two groups were analyzed by an unpaired two-tailed Student's *t* test. In the laser-CNV model, to take into account simultaneously the correlation between

measurements from the same mouse, assuming that eyes were exchangeable and the correlation for repeated measurements in the same eye (in case of repeated laser burns), a linear mixed model was used, including treatment or genotype and time as fixed effects and mouse as random effect[64,65]. Data are presented as means ± SEM, *$P < 0.05$, **$P < 0.01$, ***$P \leq 0.001$.

**Reporting summary**. Further information on research design is available in the Nature Research Reporting Summary linked to this article.

## Data availability

The authors declare that data supporting the findings of this study, including all measurement data, are available within the paper and its supplementary information files. The source data underlying Figs. 1b–d, f, h–j, l, m; 2b, c, e–g, I; 3b–d, f, h–j, l, m; 4b, c, e–g, I; 5a–d; 6a–d; 7b–d, f, h–j, l; 8b, c, e–g, I and Supplementary Figs. 1a–f; 2a, b; 3; 4a–d; 5a–f; 6b–e; 7b, d, f; 8a, b; 9a–e; 10a, b; 11a, b; 12; 13i are provided as a Source Data file.

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

## Acknowledgements

This research was supported by the Deutsche Forschungsgemeinschaft (FOR2240, project 6), the Velux Foundation, the Pro Retina Foundation, and the Erhard Rüther Foundation. We thank C. Bismar, U. Esendik, U. Karow, and E. Scheiffert for technical assistance. We also thank J. Mühlnikel, M. Retzlaff, C. Strassner, A. Jansen, and A. Lierz for support in animal caretaking. We thank Prof Marco Prinz (Institute of Neuropathology and BIOSS Center for Biological Signaling Studies, University of Freiburg, Germany) for providing Cx3cr1^CreERT2 mice.

## Author contributions

A.W. and M.H. conducted and analyzed most of the experiments. M.S. provided expertise and feedback. T.L. obtained the funding, conceived the study, and together with A.W. designed the experiments. A.W., M.H., M.S., and T.L. wrote the manuscript and all authors read and approved the final manuscript.

## Competing interests

The authors declare no competing interests.
