## [Peer Review File · Nature Communications]

Reviewers' comments:

Reviewer #1 (Remarks to the Author):

Langmann and colleagues present a superbly crafted manuscript to highlight the role of TSPO in the aetiology of the AMD. The imponent support by animal models allow them to provide a compelling set of data on the role played by the accumulation of TSPO in the development of the disease and the underlying contribution of the NOX signalling cascade.

In line with previous literature and therefore without revealing any novel mechanism _which would raise the question whether a more specialised journal would be more appropriate_ they link the pro-pathological role of TSPO in retinal degeneration with the accumulation of free species of the oxygen via the type 1 of the NADPHoxidases.

-The important pharmacological angle is nonetheless incomplete as the use of the sole XBD173 does not address the issues related with TSPO specificity of action requiring other ligands to be enrolled.

-Equally, even though well discussed, aspects on metabolism should be clarified as they would raise a lot of concerns in the community (Is the equal amount of ATP solely produced from the mitochondrial source?).

-The previous work which depicted the axis between TSPO and the Ca²⁺ dependent NOX (NOX5) cannot be here queried by the lack of this specific NOX subtype in rodents. This in turn calls for a key control which would be the actual re-insertion of the Ca²⁺ dependent NOX to address the relevance of Ca²⁺, so elegantly revealed by the authors, in the working model.

- To be better explained is also the section on extracellular ROS. If the authors are calling for a paracrine effect among cells this should be better addressed mechanistically as ROS do not travel but they just accumulate on membranes. The autocrine effect via NOX1 could/should suffice and therefore the extracellular part, which is well marked in the final model, must be detailed.

The applied value of this work remains therefore unquestionable whilst the underlying regulatory mechanisms do require further efforts in order to finalise a robust product.

Reviewer #2 (Remarks to the Author):

Langmann et al examine the role of the role of phagocytes in pathological angiogenesis in the eye and focus on the role of translocator protein (TSPO) in microglia. Using tamoxifen and CX3CR1 driven conditional deletion of TSPO in microglia, they examine the its role injury model of pathologic angiogenesis called choroidal neovascularization (CNV).

The role of microglia and monocyte derived macrophages has been extensively examined over the past 15 years and it has been demonstrated that these cells can either promote pathologic angiogenesis or prevent it depending on the microenvironmental cues and the activation state of these cells. In addition, the roles of these cells under homeostatic conditions has also been evaluated. Although there is evidence that these cells can be protective in disease models (Saban et al 2019), they can also promote neurodegeneration after activation (Wong et al, 2019 and multiple publications). As such, the nuanced understanding of the role of these cells in neurodegeneration and pathologic angiogenesis is now fairly sophisticated.

This group and others have also extensively studied the role of TSPO in microglial inflammation and phagocytosis, and in other cells such as retinal pigmented epithelium, and endothelial cells. The result of this activation has been reported to cause retinal dysfunction and neurodegeneration. In other models of cancer, TSPO has been demonstrated to regulate angiogenesis and

inflammation (Gavish et al, 2012). In age-related macular degeneration (AMD), models of which are being investigated in this study, other studies have demonstrated that TSPO regulated RPE function and lipid metabolism and as such offered TSPO targeting as a potential therapeutic option. As such, these studies represent a discovery and drug development approach but the conceptual advances are likely incremental.

Specific Comments:

Using TSPO ligands, they demonstrate that after injury, infiltration of iba1+ microglial cells was reduced in the laser CNV model (Figure 1). This is consistent with other studies where inhibitors of microglial migration such as MCP1 or other chemoattractants such as cytokines also have the same effect on microglial and monocyte burden and inflammation in laser lesions. Inhibition of TSPO also reduced ROS production from microglia which is consistent with previously published studies. VEGF is a prime driver of CNV in the eye and the investigators demonstrate a reduction of VEGF and other pro-angiogenic factors such as ang1 and 2 in the injury model. Ang1 and Ang2 are antagonistic in function and molecular elucidation of the pan-inhibitory effects of TSPO neutralization on both ang1 and ang 2 would need further investigation to determine how co-inhibition of these molecules would influence pathologic angiogenesis. In addition, ROS increase is a common factor in the effects of many of these pro-angiogenic molecules. As such, it is unclear whether the effects of TSPO are independent of the effects of VEGF.

The effects of XBD173 on microglial density within CNV lesions in most Figures is modest at best. The effects on permeability and CNV in Figure 2 are comparable to what is seen with other factors such as VEGF and Ang2. A much more detailed analysis of how TSPO interacts (or not) with VEGF and other factors in mediating the effects seen is essential given the advanced nature of the field at this age of development.

Reviewer #3 (Remarks to the Author):

This study uses multiple mouse knockout strains to demonstrate a role for TSPO in reactivity of phagocytes in a mouse model of macular degeneration. Other endpoints examined include neoangiogenesis and vascular leakage. They demonstrate that TSPO regulates Nox1 to produce reactive oxygen species in a calcium-dependent manner. Overall, the study is well done and the data are convincing. The use of knockout mice and in vitro studies strengthens the conclusions. However, in the end, the mechanistic insight is limited to showing a role for Nox1, which is already known to be calcium sensitive. Other conclusions (e.g., XBD173 inhibits inflammatory cytokines after laser injury; TSPO is induced by laser injury, XBD173 is neuroprotective) have been reported in the literature.

Specific suggestions for improvement:

1. In general, the number of replicates is not clear. One can't always tell if the retinas were from different mice (i.e., if 18 retinas were used, was this from 9 mice? If so, statistics need to be adjusted for n=9 because they are not independent samples.) or if the endpoints were from individual retinas.
2. Why does only the RPE show a 36 kD band for TSPO and retina show a 25 kD band? If 18 kD TSPO oligomerizes, it should be 36 kD or higher, yet you see a band at 25 kD. Please explain. Moreover, the physiological significance of these HMW products is unclear.
3. No statistical difference is indicated for Figure 4e, yet the text claims that CNV size was significantly reduced in TSPO microglial knockout.
4. Fig 7 purports to show reduced accumulation of Iba1+ phagocytes in the Nox1 KO, but this is not at all evident from the images. Rather, the Nox1 KO looks to have more reactivity.
5. In general, the effects of Nox1 KO are less than those of the other genetic models used. This

suggests that other pathways contribute, but this has not been considered or discussed. This is particularly evident from the fact that Nox1 is not upstream of angiogenic factors, but other endpoints are less affected as well.

Minor point

1. Fig1a is too small to see cell morphology (the main point of this panel). Same for other examples in later figures.

Reviewer #4 (Remarks to the Author):

In this interesting and very thorough study from Wolf et al, the authors provide compelling evidence that TSPO plays a role in laser-induced CNV by regulating NOX1-mediated production of ROS. The experiments are rigorous, and the results are robust and thoroughly analyzed. The findings provide significant mechanistic insight into how TSPO in microglia regulates production of ROS via NOX1, and its role in laser-induced CNV. I enjoyed the paper and have no major concerns with the conclusions.

Some questions and suggestions:

1. Is there a change in gene expression in primary microglia induced by photoreceptor debris, particularly an increase in Nox1? If so, is it dependent upon extracellular calcium? This would address whether the NOX1-dependent ROS production is in part due to increased NOX1 expression, and would also link the gene expression changes (that are dependent upon TSPO in vivo) directly to microglia and to calcium flux.

2. Line 105 states that there is "attenuated phagocyte reactivity" with XBD173 treatment and refers to Fig 1c. But this only refers to Iba1+ area. This is not a direct measure of phagocyte reactivity.

3. The sentence on line 164 is an overstatement. The word "prevented" should be "attenuated".

4. Figure 9a nicely summarizes the findings of the paper. However, this study does not assess photoreceptor degeneration, so it is misleading to include that. It would be a nice addition to assess photoreceptor degeneration in the TSPO-MG knockout and in the NOX1 knockout.

Reviewer #1 (Remarks to the Author):

Langmann and colleagues present a superbly crafted manuscript to highlight the role of TSPO in the etiology of the AMD. The imponent support by animal models allow them to provide a compelling set of data on the role played by the accumulation of TSPO in the development of the disease and the underlying contribution of the NOX signalling cascade.

In line with previous literature and therefore without revealing any novel mechanism _which would raise the question whether a more specialised journal would be more appropriate_ they link the pro-pathological role of TSPO in retinal degeneration with the accumulation of free species of the oxygen via the type 1 of the NADPH oxidases.

1. The important pharmacological angle is nonetheless incomplete as the use of the sole XBD173 does not address the issues related with TSPO specificity of action requiring other ligands to be enrolled.

Response:

We agree with the reviewer that the specificity of pharmacological effects is an important point also in the context of the TSPO ligand XBD173, which was used *in vitro* and *in vivo* in our studies.

On one hand, we believe that our genetic approach of cell-specific (Cx3cr1^{CreERT2}:TSPO^{fl/fl}) knockout of TSPO and the resulting *in vivo* analyses in the retina of these animals provide profound evidence for a specific role of TSPO in retinal phagocyte function.

On the other hand, we took the reviewer's point very serious and replicated the main mechanistic findings related to the role of TSPO in regulating Nox1-related ROS production in primary microglia using four different other TSPO ligands, including Ro5-4864, Etifoxine, PK11195 and FGIN-1-27.

As shown in the **new Supplementary Figure 3**, similarly to TSPO knockout and XBD173 treatment, these other four TSPO ligands also completely abrogated microglial ROS production induced by exposure to photoreceptor debris, indicating a specific role of TSPO in this important function.

As XBD173 offers the most beneficial side-effect profile compared to other benzodiazepine derivates (Nothdurfter et al. 2012), we believe that any TSPO-related pharmacological intervention in patients with retinal degenerative diseases would start with XBD173.

2. Equally, even though well discussed, aspects on metabolism should be clarified as they would raise a lot of concerns in the community (Is the equal amount of ATP solely produced from the mitochondrial source?).

Response:

We understand and agree with the reviewer's concern and have therefore addressed this question with further experiments. The results are shown in the **new Supplementary Figure 6**. These data show that microglia use both glycolysis and

mitochondrial respiration to generate ATP and that both mechanisms are not significantly altered in TSPO knockout microglia.

Inhibition of glycolysis through the glucose derivative 2-Deoxy-D-glucose (2-DG) showed that both non-activated and stimulated microglia to some degree depend on glycolysis for ATP generation (**new Supplementary Figure 6d**).

However, inhibition of the mitochondrial ATP synthase via oligomycin A treatment showed that ATP is mainly generated by mitochondrial respiration (**new Supplementary Figure 6e**).

We conclude that TSPO deficient microglia are fully capable of generating ATP through mitochondrial respiration further indicating unimpaired mitochondrial function.

Together with our data showing normal mitochondrial membrane potential of TSPO knockout microglia (**Supplementary Figure 6b**) and our novel data showing an unaltered mitochondrial network in TSPO knockout microglia (**new Supplementary Figure 6a**), these data support the conclusion that mitochondrial function is not altered in TSPO knockout microglia.

3. The previous work which depicted the axis between TSPO and the Ca²⁺ dependent NOX (NOX5) cannot be here queried by the lack of this specific NOX subtype in rodents. This in turn calls for a key control which would be the actual re-insertion of the Ca²⁺ dependent NOX to address the relevance of Ca²⁺, so elegantly revealed by the authors, in the working model.

Response:

We thank the reviewer for discussing this point. However, as further explained below, we refrained from re-inserting a human Nox form in the mouse model.

First, we would like to point out that it is well established that all Nox isoforms require intracellular Ca²⁺ for their activity (reviewed by Görlach et al. 2015). While activity of Nox5, Duox1 and Duox2 is **directly** regulated by cytosolic Ca²⁺, activity of the other Nox isoforms such as Nox2 or Nox1 **indirectly** depends on cytosolic Ca²⁺ levels (shown for Nox1 by Valencia et al. 2008) as the activity of the upstream regulators that are required for Nox activation such as protein kinases C depends on cytosolic Ca²⁺ levels (shown for Nox1 by Streeter et al. 2014). Furthermore, Nox1 expression levels can also depend on cytosolic Ca²⁺ (Ge et al. 2010).

The mechanisms regulating the entry of Ca²⁺ into the cytosol that is required for Nox1 activation remained elusive, though. Our data now reveal that influx of Ca²⁺ from the extracellular milieu into the cytosol is required for stimulation of Nox1 activity (Figures 6 and 7) and expression in microglia (Supplementary Figure 11) and that this influx is regulated by TSPO (Figures 6 and 7).

Thus, like Nox5 (Gatliff et al. 2017), Nox1 undoubtedly depends on the regulation of cytosolic Ca²⁺ levels by TSPO.

In our opinion, artificial introduction of Nox5 into murine microglia would not provide further insights, particularly, since Ca²⁺ regulates Nox5 and Nox1 through completely distinct mechanisms, as mentioned above.

References:

Response to reviewer's comments on NCOMMS-19-26555.

Görlach, Agnes, Katharina Bertram, Sona Hudecova, and Olga Krizanova. 2015. "Calcium and ROS: a Mutual Interplay.." *Redox Biology* 6 (December): 260–71. doi:10.1016/j.redox.2015.08.010.

Valencia, Antonio, and Irene E Kochevar. 2008. "Nox1-Based NADPH Oxidase Is the Major Source of UVA-Induced Reactive Oxygen Species in Human Keratinocytes.." *The Journal of Investigative Dermatology* 128 (1): 214–22. doi:10.1038/sj.jid.5700960.

Streeter, Jennifer, Brandon M Schickling, Shuxia Jiang, Bojana Stanic, William H Thiel, Lokesh Gakhar, Jon C D Houtman, and Francis J Miller. 2014. "Phosphorylation of Nox1 Regulates Association with NoxA1 Activation Domain.." *Circulation Research* 115 (11). Lippincott Williams & Wilkins Hagerstown, MD: 911–18. doi:10.1161/CIRCRESAHA.115.304267.

Ge, Yan, Wei Jiang, Lu Gan, Lijun Wang, Changyan Sun, Peiyan Ni, Yin Liu, et al. 2010. "Mouse Embryonic Fibroblasts From CD38 Knockout Mice Are Resistant to Oxidative Stresses Through Inhibition of Reactive Oxygen Species Production and Ca(2+) Overload.." *Biochemical and Biophysical Research Communications* 399 (2): 167–72. doi:10.1016/j.bbrc.2010.07.040.

Gatliff, Jemma, Daniel A East, Aarti Singh, Maria Soledad Alvarez, Michele Frison, Ivana Matic, Caterina Ferraina, Natalie Sampson, Federico Turkheimer, and Michelangelo Campanella. 2017. "A Role for TSPO in Mitochondrial Ca²⁺ Homeostasis and Redox Stress Signaling." *Cell Death & Disease* 8 (6). Nature Publishing Group: e2896–96. doi:10.1038/cddis.2017.186.

4. To be better explained is also the section on extracellular ROS. If the authors are calling for a paracrine effect among cells this should be better addressed mechanistically as ROS do not travel but they just accumulate on membranes. The autocrine effect via NOX1 could/should suffice and therefore the extracellular part, which is well marked in the final model, must be detailed. The applied value of this work remains therefore unquestionable whilst the underlying regulatory mechanisms do require further efforts in order to finalize a robust product.

Response:

This reviewer point is well taken and asks for further evidence that our postulated paracrine effect of extracellular ROS has a direct damaging effect on neurons.

To experimentally address this, we have used a trans-well system for co-culturing 661W photoreceptor cells with primary microglia. Flow cytometry analysis showed only a few propidium iodide-positive (PI⁺), i.e. dead, photoreceptor cells after co-culture with unstimulated microglia (**new Supplementary Figure 13**). Yet, the number of dead photoreceptor cells tremendously increased after co-culture with microglia stimulated to produce extracellular ROS through Nox1 by photoreceptor cell debris. Remarkably, scavenging of extracellular ROS with L-ascorbic acid (vitamin C) or total ROS with N-acetyl-cysteine (NAC) strongly reduced photoreceptor cell death demonstrating that ROS produced by stimulated microglia indeed induce cell death of photoreceptor cells

in a paracrine manner. Moreover, cell death of photoreceptor cells was strongly reduced if they were co-cultured with XBD173-treated microglia, microglia deficient for TSPO or microglia deficient for Nox1, respectively, which all are incapable of producing extracellular ROS through Nox1 (Figure 1m, 3m, 5c).

Of note, ROS, particularly H₂O₂, have been shown to be able to travel over short distances (Miller et al. 2009, Niethammer et al. 2009, Marinho et al. 2014, Lambeth et al. 2014). However, in these cases ROS served mainly as extracellular signaling or chemotactic molecules rather than agents that damage cells or tissues.

Therefore, the new data presented above further strengthen the conclusions of our manuscript by demonstrating that Nox1-derived extracellular ROS produced by microglia can induce photoreceptor cell death in a paracrine manner, confirming their *in vivo* potential as neurotoxins. We thank the reviewer for his comment and have modified the final model accordingly.

References:

Miller, Gad, Karen Schlauch, Rachel Tam, Diego Cortes, Miguel A Torres, Vladimir Shulaev, Jeffery L Dangl, and Ron Mittler. 2009. "The Plant NADPH Oxidase RBOHD Mediates Rapid Systemic Signaling in Response to Diverse Stimuli.." *Science Signaling* 2 (84): ra45–ra45. doi:10.1126/scisignal.2000448.

Niethammer, Philipp, Clemens Grabher, A Thomas Look, and Timothy J Mitchison. 2009. "A Tissue-Scale Gradient of Hydrogen Peroxide Mediates Rapid Wound Detection in Zebrafish.." *Nature* 459 (7249). 996–99. doi:10.1038/nature08119.

Marinho, H Susana, Carla Real, Luísa Cyrne, Helena Soares, and Fernando Antunes. 2014. "Hydrogen Peroxide Sensing, Signaling and Regulation of Transcription Factors.." *Redox Biology* 2: 535–62. doi:10.1016/j.redox.2014.02.006.

Lambeth, J David, and Andrew S Neish. 2014. "Nox Enzymes and New Thinking on Reactive Oxygen: a Double-Edged Sword Revisited." *Annual Review of Pathology: Mechanisms of Disease, Vol 9* 9 (1): 119–45. doi:10.1146/annurev-pathol-012513-104651.

Reviewer #2 (Remarks to the Author):

Langmann et al examine the role of the role of phagocytes in pathological angiogenesis in the eye and focus on the role of translocator protein (TSPO) in microglia. Using tamoxifen and CX3CR1 driven conditional deletion of TSPO in microglia, they examine the its role injury model of pathologic angiogenesis called choroidal neovascularization (CNV).

The role of microglia and monocyte derived macrophages has been extensively examined over the past 15 years and it has been demonstrated that these cells can either promote pathologic angiogenesis or prevent it depending on the microenvironmental cues and the activation state of these cells. In addition, the roles of these cells under homeostatic conditions has also been evaluated. Although there is evidence that these cells can be protective in disease models (Saban et al 2019),

they can also promote neurodegeneration after activation (Wong et al, 2019 and multiple publications). As such, the nuanced understanding of the role of these cells in neurodegeneration and pathologic angiogenesis is now fairly sophisticated.

This group and others have also extensively studied the role of TSPO in microglial inflammation and phagocytosis, and in other cells such as retinal pigmented epithelium, and endothelial cells. The result of this activation has been reported to cause retinal dysfunction and neurodegeneration. In other models of cancer, TSPO has been demonstrated to regulate angiogenesis and inflammation (Gavish et al, 2012). In age-related macular degeneration (AMD), models of which are being investigated in this study, other studies have demonstrated that TSPO regulated RPE function and lipid metabolism and as such offered TSPO targeting as a potential therapeutic option. As such, these studies represent a discovery and drug development approach, but the conceptual advances are likely incremental.

Specific Comments:

Using TSPO ligands, they demonstrate that after injury, infiltration of iba1+ microglial cells was reduced in the laser CNV model (Figure 1). This is consistent with other studies where inhibitors of microglial migration such as MCP1 or other chemoattractants such as cytokines also have the same effect on microglial and monocyte burden and inflammation in laser lesions. Inhibition of TSPO also reduced ROS production from microglia which is consistent with previously published studies. VEGF is a prime driver of CNV in the eye and the investigators demonstrate a reduction of VEGF and other pro-angiogenic factors such as ang1 and 2 in the injury model.

1. Ang1 and Ang2 are antagonistic in function and molecular elucidation of the pan-inhibitory effects of TSPO neutralization on both ang1 and ang 2 would need further investigation to determine how co-inhibition of these molecules would influence pathologic angiogenesis.

Response:

We agree with the reviewer that the consensus view of ANG-1 and ANG-2 signaling indeed is that both growth factors have opposing effects on Tie2 receptor activation since ANG-2 represents the physiological antagonist of ANG-1 and both compete for binding to the Tie2 receptor on endothelial cells. However, several studies have shown that the effects of ANG-2 can be complex and context-dependent. In the presence of high levels of VEGF, ANG-2/Tie2 signaling is pro-angiogenic leading to loss of endothelial cell junction integrity and loss of pericytes resulting in vessel sprouting. In contrast, in the absence of VEGF, ANG-2 is anti-angiogenic by inducing endothelial cell death and neovessel regression (Oshima et al. 2003; Kim et al. 2000; Bogdanovic, Nguyen, and Dumont 2006).

Since we think that an experimental approach targeting ANG-1 and ANG-2 directly in the retina is beyond the scope of this study, we have focused on a detailed experimental analysis of both factors in our experiments.

Our previous data already showed that laser damage-induced levels of VEGF, ANG-1 and ANG-2 were reduced in XBD173-treated mice and in animals with microglia-specific TSPO deficiency.

Response to reviewer's comments on NCOMMS-19-26555.

We have now performed **new experiments** (see also response to the next question and the new **Supplementary Figures 7 and 8**), which show that microglia only produce increased levels of VEGF after photoreceptor debris stimulation while ANG-1 and ANG-2 protein and mRNA levels remained unchanged and this was also independent from TSPO.

References:

Oshima, Y, S Oshima, K Takahashi, H Nambu, R S Apte, E Duh, S F Hackett, D J Zack, and P A Campochiaro. 2003. "Angiopoietin 2 (Ang2) Increases or Decreases Neovascularization (NV) Depending Upon the Setting." *Investigative Ophthalmology & Visual Science* 44 (May): U542–42.

Kim, I, J H Kim, S O Moon, H J Kwak, N G Kim, and G Y Koh. 2000. "Angiopoietin-2 at High Concentration Can Enhance Endothelial Cell Survival Through the Phosphatidylinositol 3'-Kinase/Akt Signal Transduction Pathway." *Oncogene* 19 (39). Nature Publishing Group: 4549–52. doi:10.1038/sj.onc.1203800.

Bogdanovic, Elena, Vicky P K H Nguyen, and Daniel J Dumont. 2006. "Activation of Tie2 by Angiopoietin-1 and Angiopoietin-2 Results in Their Release and Receptor Internalization." *Journal of Cell Science* 119 (Pt 17). The Company of Biologists Ltd: 3551–60. doi:10.1242/jcs.03077.

2. In addition, ROS increase is a common factor in the effects of many of these pro-angiogenic molecules. As such, it is unclear whether the effects of TSPO are independent of the effects of VEGF. The effects on permeability and CNV in Figure 2 are comparable to what is seen with other factors such as VEGF and Ang2. A much more detailed analysis of how TSPO interacts (or not) with VEGF and other factors in mediating the effects seen is essential given the advanced nature of the field at this age of development.

Response:

We agree with the reviewer that this is an important point.

To answer the question whether the effects of TSPO are independent from the effects of VEGF, ANG-1 and ANG-2, we have additionally performed *in situ* hybridization of Vegf, Ang-1 and Ang-2 together with Aif (Iba1). We showed that infiltrated Aif⁺-mononuclear phagocytes in the RPE/choroid after laser injury indeed express Vegf, Ang-1, and Ang-2 mRNA (**new Supplementary Figure 7**). In mice with microglia-specific TSPO knockout, the expression of these transcripts was diminished because of the reduced phagocyte infiltration.

For further validation and quantification of these *in situ* results, we examined mRNA expression levels of Vegf, Ang-1 and Ang-2 in isolated primary microglia using qRT-PCR. Microglia stimulated with photoreceptor cell debris showed a strong increase in Vegf transcript levels (Supplementary Figure 8a) and also increased secretion of VEGF (Supplementary Figure 8b). Ang-1 and Ang-2 expression and secretion did not change upon stimulation. Notably, XBD173 treatment or TSPO deficiency did not affect

Response to reviewer's comments on NCOMMS-19-26555.

VEGF, Ang-1 and Ang-2 expression and secretion. Thus, we conclude that the capability of microglia to produce these factors does not depend on TSPO.

3. The effects of XBD173 on microglial density within CNV lesions in most Figures is modest at best.

Response:

We agree with the reviewer that the representative images from Iba1 stains in the selected laser spots did not have the best quality. We have now increased the quality and relative size of the flat mount image subpanels in Figures 1, 3, and 7. We believe that the visual perception of the observed effect sizes now better reflects the quantitative analyses of grid cross points per cell and Iba1+ area. Given the high statistical power (see n-numbers in each subgroup), the effects of XBD173, TSPO knockout and Nox1 knockout are all highly significant.

Reviewer #3 (Remarks to the Author):

This study uses multiple mouse knockout strains to demonstrate a role for TSPO in reactivity of phagocytes in a mouse model of macular degeneration. Other endpoints examined include neoangiogenesis and vascular leakage. They demonstrate that TSPO regulates Nox1 to produce reactive oxygen species in a calcium-dependent manner.

Overall, the study is well done and the data are convincing. The use of knockout mice and in vitro studies strengthens the conclusions.

However, in the end, the mechanistic insight is limited to showing a role for Nox1, which is already known to be calcium sensitive.

Other conclusions (e.g., XBD173 inhibits inflammatory cytokines after laser injury; TSPO is induced by laser injury, XBD173 is neuroprotective) have been reported in the literature.

Response:

We thank the reviewer for appreciating the value of our work. As detailed below, we have done our best to further highlight mechanistic insights of our study following the reviewer's guidance.

For clarification, we would like to emphasize that neither the induction of TSPO itself, nor the XBD173-mediated cytokine inhibition has been published before in a laser-CNV model.

Specific suggestions for improvement:

1. In general, the number of replicates is not clear. One can't always tell if the retinas were from different mice (i.e., if 18 retinas were used, was this from 9

mice? If so, statistics need to be adjusted for n=9 because they are not independent samples.) or if the endpoints were from individual retinas.

Response:

We agree with the reviewer that the previous representation of the numbers of replicates may have been misleading. In the laser CNV-model applied here, three laser spots are individually placed concentrically around the optic nerve head in the center of the retina to both eyes of a mouse. The pharmacological treatments of the animals were systemic and the genetic manipulations also affected both eyes. According to a recent high quality publication addressing this point (Jordan 2018), two eyes can be regarded independent if the between-mouse variance is low, meaning that mice are more similar to one another than the variance in the treatment effect.

Following the reviewer's advice and where applicable, we included the term "from individual mice" in the revised Figure legends.

Reference:

Crispin Y. Jordan. Population sampling affects pseudoreplication. PLoS Biol. 2018 Oct; 16(10): e2007054. doi: 10.1371/journal.pbio.2007054

2. Why does only the RPE show a 36 kD band for TSPO and retina show a 25 kD band? If 18 kD TSPO oligomerizes, it should be 36 kD or higher, yet you see a band at 25 kD. Please explain. Moreover, the physiological significance of these HMW products is unclear.

Response:

This point of the reviewer is well taken. However, we have no definitive answer for this question so far.

TSPO expression in the RPE is constitutive and the protein is mainly present as an 18 kDa band. In the retina, the TSPO protein is also seen as a clear 18 kDa band in the non-lasered condition. In the vehicle treated, lasered group (Figure 1d), a higher molecular weight band was consistently seen at different time points (6h or 3d after laser application). XBD173 treatment completely abrogated this band. Up to this point, despite the fact that it is specific, we have no clue what this higher molecular TSPO form may do. We have never observed a higher MW form of TSPO in microglia cell lines or any other tissue.

The current understanding of the TSPO structure at molecular level is based on the NMR structure of mouse TSPO (mTSPO) (Jaremko et al. 2014, 2015) and two crystal structures of the bacterial homologs from *Rhodobacter sphaeroides* and *Bacillus cereus* (Li et al. 2015; Guo et al. 2015). Mouse TSPO was mainly reported to be monomeric in detergent systems but there are also reports indicating that a fraction of mouse TSPO may exist as oligomers in lipid bilayers (Teboul et al. 2012; Papadopoulos et al. 1994; Jaipuria et al. 2017). This suggests that different oligomeric states of TSPO may be associated with different functions. However, a recent study showed that the lipid-mimetic system which is used to solubilize mouse TSPO for NMR studies, thermodynamically destabilizes the protein, introduces structural perturbations and in addition alters the characteristics of ligand binding (Xia et al. 2019).

Response to reviewer's comments on NCOMMS-19-26555.

To address this complicated point related to the HMW TSPO band, we adapted the discussion on page 16 accordingly.

References:

Jaremko, Mariusz, Łukasz Jaremko, Garima Jaipuria, Stefan Becker, and Markus Zweckstetter. 2015. "Structure of the Mammalian TSPO/PBR Protein.." *Biochemical Society Transactions* 43 (4): 566–71. doi:10.1042/BST20150029.

Li, Fei, Jian Liu, Yi Zheng, R Michael Garavito, and Shelagh Ferguson-Miller. 2015. "Crystal Structures of Translocator Protein (TSPO) and Mutant Mimic of a Human Polymorphism." *Science* 347 (6221). American Association for the Advancement of Science: 555–58. doi:10.1126/science.1260590.

Guo, Youzhong, Ravi C Kalathur, Qun Liu, Brian Kloss, Renato Bruni, Christopher Ginter, Edda Kloppmann, Burkhard Rost, and Wayne A Hendrickson. 2015. "Structure and Activity of Tryptophan-Rich TSPO Proteins." *Science* 347 (6221): 551–55. doi:10.1126/science.aaa1534.

Teboul, David, Sylvie Beaufils, Jean-Christophe Taveau, Soria latmanen-Harbi, Anne Renault, Catherine Venien-Bryan, Veronique Vie, and Jean-Jacques Lacapère. 2012. "Mouse TSPO in a Lipid Environment Interacting with a Functionalized Monolayer." *Biochimica Et Biophysica Acta* 1818 (11): 2791–2800. doi:10.1016/j.bbamem.2012.06.020.

Papadopoulos, V, N Boujrad, M D IKONOMOVIC, P FERRARA, and B VIDIC. 1994. "Topography of the Leydig-Cell Mitochondrial Peripheral-Type Benzodiazepine Receptor." *Molecular and Cellular Endocrinology* 104 (1): R5–R9. doi:10.1016/0303-7207(94)90061-2.

Jaipuria, Garima, Andrei Leonov, Karin Giller, Suresh Kumar Vasa, Lukasz Jaremko, Mariusz Jaremko, Rasmus Linser, Stefan Becker, and Markus Zweckstetter. 2017. "Cholesterol-Mediated Allosteric Regulation of the Mitochondrial Translocator Protein Structure." *Nature Communications* 8 (1). Nature Publishing Group: –8. doi:10.1038/ncomms14893.

Xia, Yan, Kaitlyn Ledwitch, Georg Kuenze, Amanda Duran, Jun Li, Charles R Sanders, Charles Manning, and Jens Meiler. 2019. "A Unified Structural Model of the Mammalian Translocator Protein (TSPO)." *Journal of Biomolecular Nmr* 73 (6-7). Springer Netherlands: 347–64. doi:10.1007/s10858-019-00257-1.

3. No statistical difference is indicated for Figure 4e, yet the text claims that CNV size was significantly reduced in TSPO microglial knockout.

Response:

We thank the reviewer for this comment and apologize for the mistake in Figure 4e. The levels of significance were now included in the revised Figure 4e.

4. Fig 7 purports to show reduced accumulation of Iba1+ phagocytes in the Nox1 KO, but this is not at all evident from the images. Rather, the Nox1 KO looks to have more reactivity.

Response:

In the laser CNV-model, individual laser spots within one flat mount may vary to a certain extent. This can of course cause some deviations between single images. We have now done our best to better visualize Iba1-positive cells in retinal and RPE/choroidal flat mounts using our Zeiss Imager equipped with an ApoTome.2 and to select better representative images. Furthermore, the statistical analysis of several different samples allows to quantify subtle changes with high confidence. We have increased the image sizes of the subpanels a and tried to optimize the signal resolutions.

We have also uploaded higher resolution images with this revised manuscript and carefully went through all selected images. We now believe that the presented microscopic information matches relatively well with the corresponding bar chart data.

5. In general, the effects of Nox1 KO are less than those of the other genetic models used. This suggests that other pathways contribute, but this has not been considered or discussed. This is particularly evident from the fact that Nox1 is not upstream of angiogenic factors, but other endpoints are less affected as well.

Response:

We agree with the reviewer and now further discuss this important point (on page 18). Indeed, TSPO regulates microglia functions through both Nox1-dependent and Nox1-independent mechanisms. By regulating stimulus-induced Ca^{2+} influx from the extracellular milieu into the cytosol of microglia, TSPO regulates Nox1-mediated production of ROS that kill photoreceptor cells in a paracrine manner (**new Supplementary Figure 13**). TSPO-mediated production of pro-inflammatory cytokines and angiogenic factors, however, clearly is somewhat independent from Nox1. We thank the reviewer for bringing up this point and have included both the Nox1-dependent and -independent roles of TSPO in our schematic working model (Fig 9).

Minor point

6. Fig1a is too small to see cell morphology (the main point of this panel). Same for other examples in later figures.

Response:

Considering the reviewer's valid comment, we have increased the image sizes of all subpanels (a) in the respective Figures 1, 3, and 7.

Reviewer #4 (Remarks to the Author):

In this interesting and very thorough study from Wolf et al, the authors provide compelling evidence that TSPO plays a role in laser-induced CNV by regulating NOX1-mediated production of ROS. The experiments are rigorous, and the results are robust and thoroughly analyzed. The findings provide significant mechanistic insight into how TSPO in microglia regulates production of ROS via NOX1, and its role in laser-induced CNV. I enjoyed the paper and have no major concerns with the conclusions.

Some questions and suggestions:

1. Is there a change in gene expression in primary microglia induced by photoreceptor debris, particularly an increase in Nox1? If so, is it dependent upon extracellular calcium? This would address whether the NOX1-dependent ROS production is in part due to increased NOX1 expression and would also link the gene expression changes (that are dependent upon TSPO in vivo) directly to microglia and to calcium flux.

Response:

This is a very valid question. We have now analyzed the gene expression of Nox1 in stimulated and unstimulated primary microglia isolated from TSPO^{fl/fl} and TSPO^{ΔMG} mice in the presence or absence of extracellular calcium (**see new Supplementary Figure 11a, b**). We could show that Nox1 expression in WT microglia is induced by photoreceptor debris. However, upon treatment with the TSPO ligand XBD173 or in TSPO deficient microglia, this increase in Nox1 gene expression was missing (Supplementary Figure 11a, b). Interestingly, in the absence of extracellular calcium, the increase in Nox1 expression upon stimulation was also significantly reduced, suggesting that the increased ROS production indeed is in part depending on increased Nox1 expression in primary microglia.

2. Line 105 states that there is “attenuated phagocyte reactivity” with XBD173 treatment and refers to Fig 1c. But this only refers to Iba1+ area. This is not a direct measure of phagocyte reactivity.

Response:

We agree with the reviewer that referring to Figure 1c as a measurement of phagocyte reactivity was misleading. We have therefore changed the text and now refer to Figure 1b and c as the ramification assessed by the number of grid crossing points together with the Iba1-positive may better describe the (morphological) microglia reactivity.

3. The sentence on line 164 is an overstatement. The word “prevented” should be “attenuated”.

Response:

Response to reviewer's comments on NCOMMS-19-26555.

The reviewer's comment is right and we have changed the wording from "prevented" to "attenuated".

4. Figure 9a nicely summarizes the findings of the paper. However, this study does not assess photoreceptor degeneration, so it is misleading to include that. It would be a nice addition to assess photoreceptor degeneration in the TSPO-MG knockout and in the NOX1 knockout.

Response:

We agree with the reviewer that this study did not assess photoreceptor degeneration before and the schematic drawing for our final model was somehow misleading. We have now included an experiment where we analyzed photoreceptor cell death. We have used a trans-well system for co-culturing 661W photoreceptor cells with primary microglia. Flow cytometry analysis showed only a few propidium iodide-positive (PI⁺), i.e. dead, photoreceptor cells after co-culture with unstimulated microglia (**new Supplementary Figure 13**). Yet, the number of dead photoreceptor cells tremendously increased after co-culture with microglia stimulated to produce extracellular ROS through Nox1 by photoreceptor cell debris. Remarkably, scavenging of extracellular ROS with L-ascorbic acid (vitamin C) or total ROS with N-acetylcysteine (NAC) strongly reduced photoreceptor cell death demonstrating that ROS produced by stimulated microglia indeed induce cell death of photoreceptor cells in a paracrine manner. Moreover, cell death of photoreceptor cells was strongly reduced if they were co-cultured with XBD173-treated microglia, microglia deficient for TSPO or microglia deficient for Nox1, respectively, which all are incapable of producing extracellular ROS through Nox1 (Figure 1m, 3m, 5c).

This information is now also included in the adapted Figure 9.

REVIEWERS' COMMENTS:

Reviewer #1 (Remarks to the Author):

The manuscript is still lacking the sufficient mechanistic insight to clarify the intracellular pathways establishing the TSPO and NOX1. The end points are nonetheless clear-cut and rather informative. The authors have been also commendable in running extra experimental work and providing adequate explanations to the pharmacological concerns of this reviewer who shall therefore not oppose publication of the work.

Reviewer #2 (Remarks to the Author):

The authors have performed additional experimentation to address many of the reviewers comments and concerns and textually addressed some. The point that has been raised by multiple reviewers still remains which is the level of conceptual novelty that makes this study possibly a better candidate for a specialized journal as some of the pathways described have been previously characterized by this and other labs

Reviewer #3 (Remarks to the Author):

Thank you for your thoughtful response.

Reviewer #4 (Remarks to the Author):

The authors have nicely addressed the comments in the prior review. The added data strengthen the manuscript. I have no further concerns.

REVIEWERS' COMMENTS:

General comments to all reviewers

After consultation of the editor with the reviewers, a concern was raised whether the two eyes from one mouse can be regarded as independent. To account for this potential bias, we have repeated the statistical analysis of all *in vivo* laser-CNV data using a linear mixed model approach. Overall, the significance levels of the data have not changed remarkably and the conclusions were not affected. In the revised figures, the exact p-values are now given and the methods section as well as the figure legends contain further informations on the statistical test used. Thank you for bringing up this point.

Specific comments

Reviewer #1 (Remarks to the Author):

The manuscript is still lacking the sufficient mechanistic insight to clarify the intracellular pathways establishing the TSPO and NOX1. The end points are nonetheless clear-cut and rather informative.

The authors have been also commendable in running extra experimental work and providing adequate explanations to the pharmacological concerns of this reviewer who shall therefore not oppose publication of the work.

Response:

We thank the reviewer that our additional experiments and explanations were convincing and that our manuscript is supported for publication.

Reviewer #2 (Remarks to the Author):

The authors have performed additional experimentation to address many of the reviewers comments and concerns and textually addressed some. The point that has been raised by multiple reviewers still remains which is the level of conceptual novelty that makes this study possibly a better candidate for a specialized journal as some of the pathways described have been previously characterized by this and other labs

Response:

We thank the reviewer that our additional experiments find interest.

We nevertheless believe that the microglia-specific connection of TSPO-Calcium-NOX1 and retinal inflammation/angiogenesis is quite new and has not been shown before *in vivo*.

Reviewer #3 (Remarks to the Author):

Thank you for your thoughtful response.

Response:

We thank the reviewer for this supportive comment.

Reviewer #4 (Remarks to the Author):

The authors have nicely addressed the comments in the prior review. The added data strengthen the manuscript. I have no further concerns.

Response:

We thank the reviewer for this supportive comment.